# SGD with Hardness Weighted Sampling for Distributionally Robust Deep Learning

## Abstract

Distributionally Robust Optimization (DRO) has been proposed as an alternative to Empirical Risk Minimization (ERM) in order to account for potential biases in the training data distribution. However, its use in deep learning has been severely restricted due to the relative inefficiency of the optimizers available for DRO in comparison to the wide-spread Stochastic Gradient Descent (SGD) based optimizers for deep learning with ERM. In this work, we propose SGD with hardness weighted sampling, a principled and efficient optimization method for DRO in machine learning that is particularly suited in the context of deep learning. Similar to an online hard example mining strategy in essence and in practice, the proposed algorithm is straightforward to implement and computationally as efficient as SGD-based optimizers used for deep learning. It only requires adding a softmax layer and maintaining a history of the loss values for each training example to compute adaptive sampling probabilities. In contrast to typical ad hoc hard mining approaches, and exploiting recent theoretical results in deep learning optimization, we prove the convergence of our DRO algorithm for over-parameterized deep learning networks with ReLU activation and finite number of layers and parameters. Preliminary results demonstrate the feasibility and usefulness of our approach.

## 1 Introduction

In standard deep learning pipelines, a neural network $h$ with parameters $\boldsymbol{\theta}$ is trained by minimizing the mean of a per-example loss $\mathcal{L}$ over a training dataset $\{(\mathrm{x}_i, \mathrm{y}_i)\}_{i=1}^{n}$, where $\mathrm{x}_i$ are the inputs and $\boldsymbol{y}_i$ are the labels. This corresponds to Empirical Risk Minimization (ERM), defined as the non-convex optimization problem

$$\arg\min_{\boldsymbol{\theta}} \frac{1}{n} \sum_{i=1}^{n} \mathcal{L}\left(h(\mathrm{x}_i; \boldsymbol{\theta}), \mathrm{y}_i\right) \tag{1}$$

Since the empirical risk is equal to the expectation of the per-example loss over the empirical training data distribution, an approximate solution of (1) can be obtained efficiently by Stochastic Gradient Descent (SGD) with a uniform sampling over the training data (Bottou et al., 2018).

This approach has led to remarkable results in terms of average performance, but may lead to outliers with high loss values compared to the average loss. Such cases can even be observed for elements belonging to the training data set. This is because solutions of ERM are prone to ignore a few hard examples in order to obtain a low mean per-example loss.

In practice, outlier results have, for example, been consistently reported in the context of deep learning for brain tumor segmentation, as illustrated in the recent annual BRATS challenges (Bakas et al., 2018). For safety-critical systems, such as those used in healthcare, where outliers must be avoided, this is not satisfactory.

Efficient biased sampling methods, including Online Hard Example Mining (OHEM) (Shrivastava et al., 2016; Loshchilov & Hutter, 2015; Chang et al., 2017) and weighted sampling (Bouchard et al., 2015; Berger et al., 2018; Gibson et al., 2018), have been proposed to mitigate this issue. However, even though these works typically start from an ERM formulation, it is not clear how those heuristics actually relate to ERM in theory.

Distributionally Robust Optimization (DRO) is an alternative to ERM (1) that takes into account uncertainty in the empirical data distribution. Formally, training a deep neural network with DRO corresponds to the min-max non-convex-concave optimization problem

$$\arg\min_{\boldsymbol{\theta}} \max_{q} \left( \sum_{i=1}^{n} q_i \, \mathcal{L}\left(h(\mathbf{x}_i; \boldsymbol{\theta}), \mathbf{y}_i\right) - \frac{1}{\beta} \sum_{i=1}^{n} \frac{1}{n} \phi\left(nq_i\right) \right) \tag{2}$$

where $\phi$ is a convex function that defines a $\phi$-divergence (Csiszár et al., 2004), $q = (q_i)_{i=1}^{n}$ corresponds to an arbitrary sampling distribution over the training data, and $\beta > 0$ is a robustness parameter. Instead of minimizing the mean per-example loss on the training dataset, DRO seeks the hardest *weighted* empirical training data distribution around the (uniform) empirical training data distribution. This suggests a link between DRO and OHEM.

The parameter $\beta$ allows DRO to interpolate between ERM ($\beta \rightarrow 0$) and the minimization of the maximum per-example loss ($\beta \rightarrow +\infty$). Motivations for using the minimization of the maximum per-example loss for safety-critical applications have been discussed in (Shalev-Shwartz & Wexler, 2016).

DRO as a generalization of ERM for machine learning has been studied in (Duchi et al., 2016; Rafique et al., 2018; Namkoong & Duchi, 2016; Chouzenoux et al., 2019), but still lacks optimization methods that are computationally as efficient as SGD in the non-convex setting of deep learning.

If one could efficiently solve the inner maximization problem in (2) for a given $\boldsymbol{\theta}$, DRO could be addressed by alternating between this maximization problem and a minimization scheme akin to the standard ERM (1), but over an adaptively weighted empirical distribution. However, even when a closed-form solution is available for the inner maximization problem, it requires performing a forward pass over the entire training dataset at each iteration. This cannot be done efficiently for large dataset.

Previously proposed optimization methods for large-scale non-convex-concave problem of the form of (2) are based on the min-max structure of the problem, and consist in alternating between approximate maximization and minimization steps (Rafique et al., 2018; Lin et al., 2019; Jin et al., 2019). However, they differ from SGD methods for ERM by the introduction of additional hyperparameters for the optimizer such as a second learning rate and a ratio between the number of minimization and maximization steps. As a result, DRO is difficult to use as a replacement of ERM in practice. In addition, those min-max methods do not use the link between DRO and adaptive weighted sampling, therefore departing from efficient heuristics used in hard example mining. From a theoretical perspective, they further make the assumption that the model is either smooth or weakly-convex, but none of those properties are true for deep neural networks with ReLU activation functions that are largely used in practice.

In this work, we propose SGD with hardness weighted sampling, a novel, principled optimization method for training deep neural networks with DRO (2) and inspired by OHEM. Compared to SGD, our method only requires introducing an additional $\mathrm{softmax}$ layer and maintaining a history of the stale per-example loss to compute sampling probabilities over the training data. Since the loss is already computed at each iteration for SGD, our SGD with hardness weighted sampling for DRO is computationally as efficient as SGD for ERM. In practice, we show that our method performs favorably to SGD in the case of class imbalance.

We also formally link DRO in our method with OHEM (Shrivastava et al., 2016). As a result, our method can be seen as a principled OHEM approach. In this context, the robustness parameter $\beta$ controls the trade-off between exploitation and exploration in the OHEM process.

Last but not least, we generalize recent results in the convergence theory of SGD with ERM and over-parameterized deep learning networks with ReLU activation functions (Allen-Zhu et al., 2019; 2018; Cao & Gu, 2019; Zou & Gu, 2019) to our SGD with hardness weighted sampling for DRO. This is, to the best of our knowledge, the first convergence result for deep learning networks with ReLU trained with DRO.

## 2 RELATED WORK IN DRO WITH A WASSERSTEIN DISTANCE

In this work, we focus on DRO with a $\phi$-divergence (Csiszár et al., 2004). In this case, the data distributions that are considered in the DRO problem (2) are restricted to sharing the support of the empirical training distribution. In other words, the weights assigned to the training data can change, but the training data itself remains unchanged.

Another popular formulation is DRO with a Wasserstein distance (Sinha et al., 2017; Duchi et al., 2016; Staib & Jegelka, 2017; Chouzenoux et al., 2019). In contrast to $\phi$-divergences, using a Wasserstein distance in DRO seeks to apply small data augmentation to the training data to make the deep learning model robust to small deformation of the data, but the sampling weights of the training data distribution typically remains unchanged. In this sense, DRO with a $\phi$-divergence and DRO with a Wasserstein distance can be considered as orthogonal endeavours.

While we show that DRO with $\phi$-divergence can be seen as a principled OHEM method, it has been shown that DRO with a Wasserstein distance can be seen as a principled adversarial training method (Sinha et al., 2017; Staib & Jegelka, 2017).

## 3 MACHINE LEARNING WITH DRO AND $\phi$-DIVERGENCE

In machine learning based on Empirical Risk Minimization (ERM), a predictor $h$ is trained using a training dataset $\{(\boldsymbol{x}_i, \boldsymbol{y}_i)\}_{i=1}^n$ to perform well on average on a task for which the performance is measured on a per-example basis by a smooth criteria $\mathcal{L}$. Note that parameter regularization terms can easily be embedded in $\mathcal{L}$ since they are independent of the example. For ease of presentation, we focus on the supervised machine learning setting, where $h : \boldsymbol{x} \mapsto \boldsymbol{y}$, and omit explicitly mentioning any parameter regularisation term.

Let $\Delta_n \subset \mathbb{R}^n$ be the set of empirical weighted training data distributions defined according to a given training dataset

$$\Delta_n = \left\{ p = (p_i)_{i=1}^n \in [0, 1]^n \mid \sum_{i=1}^n p_i = 1 \right\} \tag{3}$$

and let $\hat{p}_{\text{data}}$ be the uniform empirical training data distribution, i.e. for all $i$, $\hat{p}_{\text{data},i} = \frac{1}{n}$.

Let $\boldsymbol{\theta}$ be the set of parameters of the predictor $h(\,.\,;\boldsymbol{\theta}) : \boldsymbol{x} \mapsto \boldsymbol{y}$ we want to train, and $\boldsymbol{h} : \boldsymbol{\theta} \mapsto (h(\boldsymbol{x}_i; \boldsymbol{\theta}))_{i=1}^n$ be the vector of inferred outputs from the training data. We assume $\mathcal{L}$ is a smooth and potentially non-convex function. We also denote $\mathcal{L}(\boldsymbol{h}(\boldsymbol{\theta})) = (\mathcal{L}(h(\boldsymbol{x}_i; \boldsymbol{\theta}), \boldsymbol{y}_i))_{i=1}^n$.

**Definition 3.1** (Mean Loss)**.**

$$M(\mathcal{L}(\boldsymbol{h}(\boldsymbol{\theta}))) = \mathbb{E}_{\hat{p}_{\text{data}}} [\mathcal{L}(h(\mathrm{x}; \boldsymbol{\theta}), \boldsymbol{y})] = \frac{1}{n} \sum_{i=1}^n \mathcal{L}(h(\mathrm{x}_i; \boldsymbol{\theta}), \boldsymbol{y}_i) \tag{4}$$

The ERM predictor, as used in most learning settings, is obtained by minimizing the mean loss (4).

**Definition 3.2** (Empirical Risk Minimization (ERM) predictor)**.**

$$\tilde{\boldsymbol{\theta}} = \arg\min_{\boldsymbol{\theta}} M(\mathcal{L}(\boldsymbol{h}(\boldsymbol{\theta}))) \tag{5}$$

However, $\hat{p}_{\text{data}}$ is typically biased compared to the true data distribution. Therefore, predictors trained with ERM are prone to fail on new examples that are not well represented in the training dataset.

Distributionally Robust Optimization (DRO) is an alternative to ERM that mitigates this issue by encouraging robustness to bias in the empirical training data distribution. DRO, in its simplest form, is based on the notion of $\phi$-divergence that we use to induce robustness with respect to the set of all the empirical distributions of the training dataset $\Delta_n$.

**Definition 3.3** ($\phi$-divergence)**.** *Let $\phi : \mathbb{R}_+ \to \mathbb{R} \cup \{+\infty\}$ be a closed, convex, lower semi-continuous function such that $\forall z \in \mathbb{R}_+, \phi(z) \geq \phi(1) = 0$. The $\phi$-divergence $D_\phi$ is defined as, for all $p = (p_i)_{i=1}^n, q = (q_i)_{i=1}^n \in \Delta_n$*

$$D_\phi(q\|p) = \sum_{i=1}^n p_i \phi\left(\frac{q_i}{p_i}\right) \tag{6}$$

**Example 3.1.** *For $\phi : z \mapsto z \log(z)$, $D_\phi$ is the Kullback-Leibler (KL) divergence:*

$$D_\phi(q\|p) = D_{\mathrm{KL}}(q\|p) = \sum_{i=1}^{n} q_i \log\left(\frac{q_i}{p_i}\right) \tag{7}$$

*And, for $\phi : z \mapsto (z-1)^2$, $D_\phi$ is the Pearson $\chi^2$ divergence:*

$$D_\phi(q\|p) = \chi^2(q\|p) = \sum_{i=1}^{n} \frac{(q_i - p_i)^2}{p_i} \tag{8}$$

**Definition 3.4** (Distributionally Robust Loss).

$$\begin{aligned} R(\mathcal{L}(\boldsymbol{h}(\boldsymbol{\theta}))) &= \max_{q\in\Delta_n} \mathbb{E}_q\left[\mathcal{L}\left(h(\mathrm{x};\boldsymbol{\theta}),\boldsymbol{y}\right)\right] - \frac{1}{\beta}D_\phi(q\|\hat{p}_{\mathrm{data}}) \\ &= \max_{q\in\Delta_n} \sum_{i=1}^{n} q_i\,\mathcal{L}\left(h(\mathrm{x}_i;\boldsymbol{\theta}),\boldsymbol{y}_i\right) - \frac{1}{n\beta}\sum_{i=1}^{n}\phi\left(nq_i\right) \end{aligned} \tag{9}$$

*where $\beta > 0$ is a hyperparameter that controls the amount of robustness.*

For a given $\phi$-divergence, we define the DRO predictor, that is obtained by minimizing the distributionally robust loss (9) instead of the mean loss (4).

**Definition 3.5** (Distributionally Robust Optimization (DRO) predictor).

$$\tilde{\boldsymbol{\theta}} = \arg\min_{\boldsymbol{\theta}} R(\mathcal{L}(\boldsymbol{h}(\boldsymbol{\theta}))) \tag{10}$$

DRO interpolates between ERM as $\beta \to 0$ and the minimization of the maximum loss as $\beta \to \infty$, and is equivalent to a mean-variance trade-off when $\beta \to 0$ small (Gotoh et al., 2018), i.e.

$$\begin{cases} \max_{q\in\Delta_n}\left(\mathbb{E}_q\left[\mathcal{L}\left(h\left(\mathrm{x};\boldsymbol{\theta}\right),\mathrm{y}\right)\right] - \frac{1}{\beta}D_\phi(q\|\hat{p}_{\mathrm{data}})\right) = \mathbb{E}_{\hat{p}_{\mathrm{data}}}\left[\mathcal{L}(h\left(\mathrm{x};\boldsymbol{\theta}\right),\mathrm{y})\right] \\ \qquad\qquad\qquad\qquad\qquad\qquad\qquad + \frac{\beta}{2\phi''(1)}\mathbb{V}_{\hat{p}_{\mathrm{data}}}\left[\mathcal{L}(h\left(\mathrm{x};\boldsymbol{\theta}\right),\boldsymbol{y})\right] + o(\beta) \quad (11) \\ \max_{q\in\Delta_n}\left(\mathbb{E}_q\left[\mathcal{L}\left(h\left(\mathrm{x};\boldsymbol{\theta}\right),\mathrm{y}\right)\right] - \frac{1}{\beta}D_\phi(q\|\hat{p}_{\mathrm{data}})\right) \xrightarrow[\beta\to+\infty]{} \max_i \mathcal{L}(h\left(\mathrm{x}_i;\boldsymbol{\theta}\right),\mathrm{y}_i) \end{cases}$$

where $\mathbb{V}_{\hat{p}_{\mathrm{data}}}$ is the empirical variance.

Furthermore, we observe that the distributionally robust loss (9) is an upper bound to the mean loss (4) (independently to the choice of $\phi$ and $\beta$), i.e. for all $\phi$-divergence and all $\beta > 0$

$$\forall\boldsymbol{\theta}, \quad M(\mathcal{L}(\boldsymbol{h}(\boldsymbol{\theta}))) \le R(\mathcal{L}(\boldsymbol{h}(\boldsymbol{\theta}))) \tag{12}$$

We now make assumptions for the $\phi$-divergence to simplify the derivations of our optimization method for DRO (10) in the remainder of the paper.

**Assumption 3.1** (Regularity of $\phi$). *$\phi : \mathbb{R}_+ \to \mathbb{R}$ is two times continuously differentiable on $[0,n]$, $\rho$-strongly convex on $[0,n]$, i.e.: $\exists\rho > 0, \forall z,z' \in [0,n], \; \phi(z') \ge \phi(z) + \phi'(z)(z'-z) + \frac{\rho}{2}(z-z')^2$ and satisfies (see D.1 for a justification): $\forall z \in \mathbb{R}, \; \phi(z) \ge \phi(1) = 0, \; \phi'(1) = 0$.*

These assumptions are verified by most $\phi$-divergences used in practice (e.g. the KL divergence).

## 4 SGD WITH HARDNESS WEIGHTED SAMPLING

### 4.1 DISTRIBUTIONALLY ROBUST OPTIMIZATION WITH SGD AND ADAPTIVE SAMPLING

Existing optimization methods for DRO with a non-convex predictor $h$ alternate between approximate minimization and maximization steps (Rafique et al., 2018; Jin et al., 2019; Lin et al., 2019), requiring the introduction of additional hyperparameters compared to SGD. These are difficult to

tune in practice and convergence has not been proven for deep neural networks with ReLU activation functions.

In this section, we highlight properties that allows us to link DRO with SGD combined with adaptive sampling. Our analysis relies on Fenchel duality (Moreau, 1965) and the notion of Fenchel conjugate (Fenchel, 1949) that we now define.

**Definition 4.1** (Fenchel Conjugate Function). *Let $f : \mathbb{R}^m \to \mathbb{R} \cup \{+\infty\}$ be a proper function. The Fenchel conjugate of $f$ is defined as $\forall \boldsymbol{v} \in \mathbb{R}^m, \; f^*(\boldsymbol{v}) = \max_{\boldsymbol{x} \in \mathbb{R}^m} \langle \boldsymbol{v}, \boldsymbol{x} \rangle - f(\boldsymbol{x})$.*

Let

$$\forall p \in \mathbb{R}^n, \quad G(p) = \frac{1}{\beta} D_\phi(p \| p_{train}) + \delta_{\Delta_n}(p) \tag{13}$$

where $\delta_{\Delta_n}$ is the characteristic function of the closed convex set $\Delta_n$, i.e.

$$\forall p \in \mathbb{R}^n, \quad \delta_{\Delta_n}(p) = \begin{cases} 0 & \text{if } p \in \Delta_n \\ +\infty & \text{otherwise} \end{cases} \tag{14}$$

One can remark that the distributionally robust loss $R$ (9) can be rewritten using the Fenchel conjugate function of $G$. This allows us to obtain regularity properties for $R$.

**Lemma 4.1** (Regularity of $R$). *If $\phi$ satisfies Assumption 3.1, then $G$ and $R$ satisfy the following:*

$$G \text{ is } \left( \frac{n\rho}{\beta} \right) \text{-strongly convex} \tag{15}$$

$$\forall \boldsymbol{\theta}, \quad R(\mathcal{L}(h(\boldsymbol{\theta}))) = \max_{q \in \mathbb{R}^n} \left( \langle \mathcal{L}(h(\boldsymbol{\theta})), q \rangle - G(q) \right) = G^* \left( \mathcal{L}(h(\boldsymbol{\theta})) \right) \tag{16}$$

$$R \text{ is } \left( \frac{\beta}{n\rho} \right) \text{-gradient Lipschitz continuous.} \tag{17}$$

Equation (16) follows from Definition 4.1. Proofs of (15) and (17) can be found in Appendix D.3.

According to (15), the optimization problem (16) is strictly convex and admit a unique solution in $\Delta_n$. Let us denote this solution

$$\bar{p}(\mathcal{L}(h(\boldsymbol{\theta}))) = \arg\max_{q \in \mathbb{R}^n} \left( \langle \mathcal{L}(h(\boldsymbol{\theta})), q \rangle - G(q) \right) \tag{18}$$

The following lemma shows that the gradient, with respect to $\boldsymbol{\theta}$, of the distributionally robust loss (9) at a given $\boldsymbol{\theta}$ can be rewritten as the expectation, with respect to the weighted empirical distribution $\bar{p}(\mathcal{L}(h(\boldsymbol{\theta})))$, of the per-example loss gradient. We further show that straightforward analytical formulas exist for $\bar{p}$ when relying on classical $\phi$-divergences. This result motivates our Algorithm 4.1 for efficient training with the distributionally robust loss.

**Lemma 4.2** (Stochastic Gradient of the Distributionally Robust Loss). *For all $\boldsymbol{\theta}$, we have*

$$\begin{aligned} \bar{p}(\mathcal{L}(h(\boldsymbol{\theta}))) &= \nabla_{\boldsymbol{v}} R(\mathcal{L}(h(\boldsymbol{\theta}))) \\ \nabla_{\boldsymbol{\theta}} (R \circ \mathcal{L} \circ h)(\boldsymbol{\theta}) &= \mathbb{E}_{\bar{p}(\mathcal{L}(h(\boldsymbol{\theta})))} \left[ \nabla_{\boldsymbol{\theta}} \mathcal{L}(h(\mathrm{x}; \boldsymbol{\theta}), y) \right] \end{aligned} \tag{19}$$

*where $\nabla_{\boldsymbol{v}} R$ is the gradient of $R$ with respect to its input.*

The proof is found in Appendix D.4. It is apparent from (19) that, given $\bar{p}$, an estimate of $\nabla_{\boldsymbol{\theta}} (R \circ \mathcal{L} \circ h)$ could easily be provided by sampling a batch according to $\bar{p}$ and estimating the per-example loss gradients in the batch as per standard practice. We now provide closed-form formulas for $\bar{p}$ given $\mathcal{L}(h(\boldsymbol{\theta}))$ for the KL divergence and the Pearson $\chi^2$ divergence.

**Example 4.1.** *For the KL divergence (i.e. $\phi : z \mapsto z \log(z) - z + 1$), we have (see D.2 for a proof)*

$$\bar{p}(\mathcal{L}(h(\boldsymbol{\theta}))) = \text{softmax}\left( \beta \, \mathcal{L}(h(\boldsymbol{\theta})) \right) \tag{20}$$

*And for the Pearson $\chi^2$ divergence (i.e. $\phi : z \mapsto (z - 1)^2$), we have:*

$$\forall i, \quad \bar{p}_i(\mathcal{L}(h(\boldsymbol{\theta}))) = \text{ReLU}\left( \frac{1}{n} \left( 1 + \frac{\beta}{2} \left( \mathcal{L}(h(\boldsymbol{\theta}))_i - \frac{1}{n} \sum_{j=1}^n \mathcal{L}(h(\boldsymbol{\theta}))_j \right) \right) \right) \tag{21}$$

*In both cases, we can verify consistency with (11) as*

$$\forall i \in \{1, \dots, n\}, \quad \bar{p}_i(\mathcal{L}(h(\boldsymbol{\theta}))) \xrightarrow[\beta \to 0]{} \frac{1}{n} \tag{22}$$

---

**Algorithm 1** SGD-HWS: SGD with Hardness Weighted Sampling for Kullback-Leibler DRO

---

1: **Input:** Training data $\{(x_i, y_i)\}_{i=1}^n$, number of epochs $T > 1$, robustness parameter $\beta > 0$, learning rate $\eta > 0$, batch size $b \in \{1, \ldots, n\}$.
2: **Initialization:**
3: Initialise $\boldsymbol{\theta}$ randomly
4: Initialise the loss history $\tilde{\mathcal{L}} = -1$
5: **Warm start**:
6:     // Split the training data into batches $\mathcal{B}$ and run one epoch with classic SGD
7: **for** $\{(x_i, y_i)\}_{i \in I}$ in $\mathcal{B}$ **do**
8:     // Run forward pass and store losses for all the samples in the batch
9:     **for** $i \in I$ **do**
10:         $\tilde{\mathcal{L}}_i \leftarrow \mathcal{L}(h(x_i; \boldsymbol{\theta}), y_i)$
11:     // Run backward pass and update the parameters of the model
12:     $\boldsymbol{\theta} \leftarrow \boldsymbol{\theta} - \eta \frac{1}{b} \sum_{i \in I} \nabla_{\boldsymbol{\theta}} \mathcal{L}(h(x_i; \boldsymbol{\theta}), y_i)$
13: **SGD with dynamic hardness weighted sampling:**
14: **for** epoch $= 2, \ldots, T$ **do**
15:     **for** iteration $i = 1, \ldots, \left( \left\lfloor \frac{n}{b} \right\rfloor + 1 \right)$ **do**
16:         // Run softmax to update the sampling probabilities of the samples
17:         $\hat{p} = \text{softmax}(\beta \tilde{\mathcal{L}})$
18:         // Draw a batch with replacement using the probability distribution $\hat{p}$
19:         $\{(x_i, y_i)\}_{i \in I}$ such that $I \overset{\text{i.i.d.}}{\sim} \tilde{p}$ and $|I| = b$
20:         // Run forward pass and update losses for all the samples in the batch
21:         **for** $i \in I$ **do**
22:             $\tilde{\mathcal{L}}_i \leftarrow \mathcal{L}(h(x_i; \boldsymbol{\theta}), y_i)$
23:         // Run backward pass and update the parameters of the model
24:         $\boldsymbol{\theta} \leftarrow \boldsymbol{\theta} - \eta \frac{1}{b} \sum_{i \in I} \nabla_{\boldsymbol{\theta}} \mathcal{L}(h(x_i; \boldsymbol{\theta}), y_i)$
25: **Output:** $\theta$

---

## 4.2 EFFICIENT ALGORITHM FOR DISTRIBUTIONALLY ROBUST DEEP LEARNING

The second equality in (19) implies that $\nabla_{\boldsymbol{\theta}} \mathcal{L}(h_i(\boldsymbol{\theta}), \boldsymbol{y}_i)$ is an unbiased estimator of the distributionally robust loss gradient when $i$ is sampled with respect to $\bar{p}(\mathcal{L}(h(\boldsymbol{\theta})))$. This suggests that the distributionally robust loss can be minimized efficiently by SGD by sampling mini-batches with respect to $\bar{p}(\mathcal{L}(h(\boldsymbol{\theta})))$ at each iteration. However, even though closed-form formulas were provided for $\bar{p}$, evaluating exactly $\mathcal{L}(h(\boldsymbol{\theta}))$, i.e. doing one forward pass on the whole training dataset at each iteration, is computationally prohibitive for large training dataset.

In practice, we propose to use a stale version of $\mathcal{L}(h(\boldsymbol{\theta}))$ by maintaining an online history of the loss values of the training examples during training $\big(\mathcal{L}(h(\boldsymbol{x}_i; \boldsymbol{\theta}^{(t_i)}), \boldsymbol{y}_i)\big)$. Where for all $i$, $t_i$ is the last iteration at which the per-example loss of example $i$ has been computed. Using the Kullback-Leibler divergence as $\phi$-divergence, this leads to the SGD with hardness weighted sampling algorithm proposed in Algorithm 4.1. This would also apply to the Pearson $\chi^2$ divergence mutatis mutandis.

In contrast to alternate min-max optimization methods, our SGD with an adaptive sampling strategy is similar to the SGD-based optimizers used by the vast majority of deep learning practitioners. Compared to standard SGD-based training optimizers for the mean loss, our algorithm requires only an additional softmax operation per iteration and to store an additional vector of size $n$ (number of training examples), thereby making it ideally suited for deep learning applications.

## 4.3 CONVERGENCE OF SGD WITH HARDNESS WEIGHTED SAMPLING FOR OVER-PARAMETERIZED DEEP NEURAL NETWORKS WITH ReLU

Convergence results for over-parameterized deep learning has recently been proposed in (Allen-Zhu et al., 2019). Their work gives convergence guarantees for deep neural networks $h$ with any activation functions (including ReLU), and with any (finite) number of layers $L$ and parameters $m$,

under the assumption that $m$ is large enough. At the time of writing, this is the most realistic setting for which a convergence theory of deep learning exists.

In this section, we extend the convergence theory developed by (Allen-Zhu et al., 2019) for ERM and SGD to DRO and the proposed SGD with hardness weighted sampling (as stated in Algorithm 4.1).

**Theorem 4.1** (Convergence of Algorithm 4.1 for over-parameterized neural networks with $\mathrm{ReLU}$). *Let $\mathcal{L}$ be a smooth per-sample loss function, $b \in \{1, \ldots, n\}$ be the batch size, and $\epsilon > 0$. If $m$ is large enough, and the learning rate is small enough, then, with high probability over the randomness of the initialization and the mini-batches, Algorithm 4.1 finds $\|\nabla_{\boldsymbol{\theta}}(R \circ \mathcal{L} \circ \boldsymbol{h})(\boldsymbol{\theta})\| \leq \epsilon$ after a finite number of iterations.*

A more detailed version of this theorem is described in B.2 and the proof can be found in D.8.

## 4.4 DRO as Principled Online Hard Example Mining

In this section, we discuss the relationship between DRO and Online Hard Example Mining (OHEM) (Shrivastava et al., 2016). SGD with an ad hoc adaptive sampling strategy is already used in practice while starting from a mean loss optimization formulation in the OHEM literature (Loshchilov & Hutter, 2015; Shrivastava et al., 2016). Similarly to our algorithm, in OHEM heuristics, the *hard examples*, those training examples with relatively high values of the loss, are sampled more often. We formalize this in the following definition for OHEM sampling.

**Definition 4.2** (Online Hard Example Mining Sampling). *Any adaptive sampling method such that the probability $p_i$ of sampling example $\mathrm{x}_i$ is an non-decreasing function of the (potentially stale) loss value associated with $\mathrm{x}_i$.*

**Theorem 4.2.** *The proposed hardness weighted sampling is a hard example mining sampling for any $\phi$-divergence that satisfies Assumption 3.1. In addition, the probability $p_i$ of sampling example $\mathrm{x}_i$ is an non-increasing function of the loss value associated with $\mathrm{x}_j$ for all $j \neq i$.*

See Appendix D.5 for the proof. The second part of Theorem 4.2 implies that as the loss of an example diminishes, the sampling probabilities of all the other examples increase. As a result, the proposed SGD with hardness weighted sampling balances exploitation (i.e. sampling the identified *hard examples*) and exploration (i.e. sampling any example to keep the record of *hard examples* up to date).

# 5 Experiments

We now illustrate the properties of our SGD with hardness weighted sampling described in Algorithm 4.1 for training deep neural networks with ReLU activation functions for DRO (10).

## 5.1 Robustness to domain gap

We create a bias between training and testing data distribution of MNIST (LeCun, 1998) by keeping only $1\%$ of the digits 3 in the training dataset, while the testing dataset remains unchanged. Implementation details can be found in Appendix A.1.

Comparison of ERM with SGD and DRO with our SGD with hardness weighted sampling for $\beta = 10$ at testing can be found in Figure 1. More values of $\beta$ and the learning curves during training can be found in Figure 2.

Our experiment suggests that DRO and ERM lead to different optima. Indeed, DRO for $\beta = 10$ outperforms ERM by more than $10\%$ of accuracy on the under-represented class, as illustrated in Figure 1. This suggests that DRO leads to better generalization than ERM. Especially, it appears that DRO is more robust than ERM to domain gaps between the training and the testing dataset. In addition, Figure 1 suggests that DRO with our SGD with hardness weighted sampling can convergence faster than ERM with SGD.

Furthermore, the variations of learning curves with $\beta$ that can be found in Figure 2 are consistent with our theoretical insight. As $\beta$ decreases to 0, the learning curve of DRO with our Algorithm 4.1 converges to the learning curve of ERM with SGD.

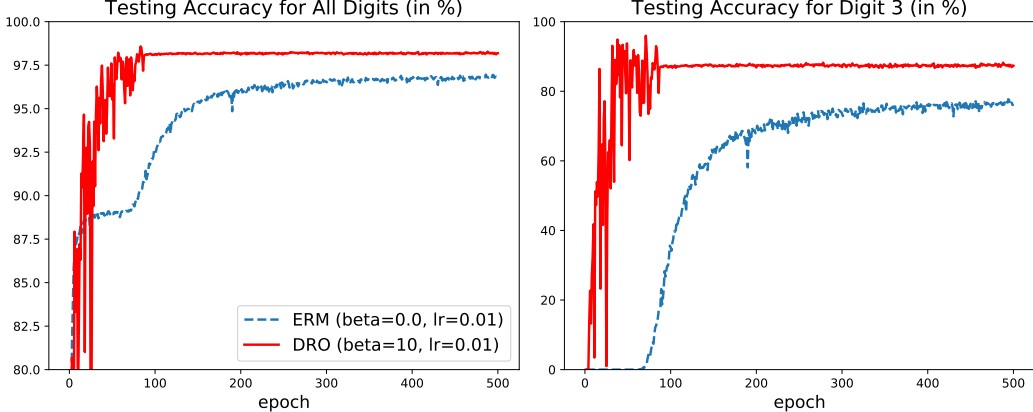

Figure 1: Comparison of learning curves for ERM with SGD (blue) and DRO with our SGD with hardness weighted sampling (red). The models are trained on an imbalanced MNIST dataset (only 1% of the digits 3 kept in the training datatset) and evaluated on the original MNIST testing dataset. This suggests that our SGD with hardness weighted sampling is more robust than SGD to a domain gap between the training and the testing dataset.

## 5.2 Stability of DRO

For large values of $\beta$ (here $\beta \geq 10$), instabilities appear in the **testing learning curves** at the begining of training, as illustrated in Figure 1 and in the top panels of Figure 2. For ERM this usually suggests that the learning rate is too high.

However, we observed that reducing the learning rate does not reduce those instabilities. The bottom left panel of Figure 2 shows that the **training loss curves** for $\beta \geq 10$ were actually stable there. We also observe that during the iterations for which instabilities appear on the **testing set**, the standard deviation of the per-sample loss on the **training set** increases and then decreases. Following (20) the higher the standard deviation of the per-sample loss history, the more our weighted sampler focuses on *hard examples*. Therefore, instabilities on the **testing set** during training with DRO are due to the sampler focusing on *hard examples*.

## 6 Conclusion and Discussion

We have shown that efficient training of deep neural networks with Distributionally Robust Optimization (DRO) with a $\phi$-divergence (10) is possible. Our Stochastic Gradient Descent (SGD) with hardness weighted sampling is a principled Online Hard Example Mining (OHEM) method. It is as straightforward to implement, and as computationally efficient as SGD for Empirical Risk Minimization (ERM). It can be used for deep neural networks with any activation function (including ReLU), and with any per-example loss function. We have shown that the proposed approach can formally be described as a principled online hard example mining strategy. In addition, we prove the convergence of our method for over-parameterized deep neural networks. Thereby, extending the convergence theory of deep learning of (Allen-Zhu et al., 2019). This is, to the best of our knowledge, the first convergence result for training a deep neural network based on DRO.

Our experiments on an imbalanced MNIST dataset illustrate the practical usefulness and feasibility of our methods. SGD with hardness weighted sampling is more robust to domain gaps between the training and the testing dataset and converges faster than SGD.

However, adapting acceleration methods for ERM with SGD, like momentum updates, to DRO remains non-trivial because of the inner maximization of DRO. Investigating accelerated extension of our method is left for future work.

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

# A MORE ON OUR EXPERIMENTS

## A.1 EXPERIMENTS ON MNIST

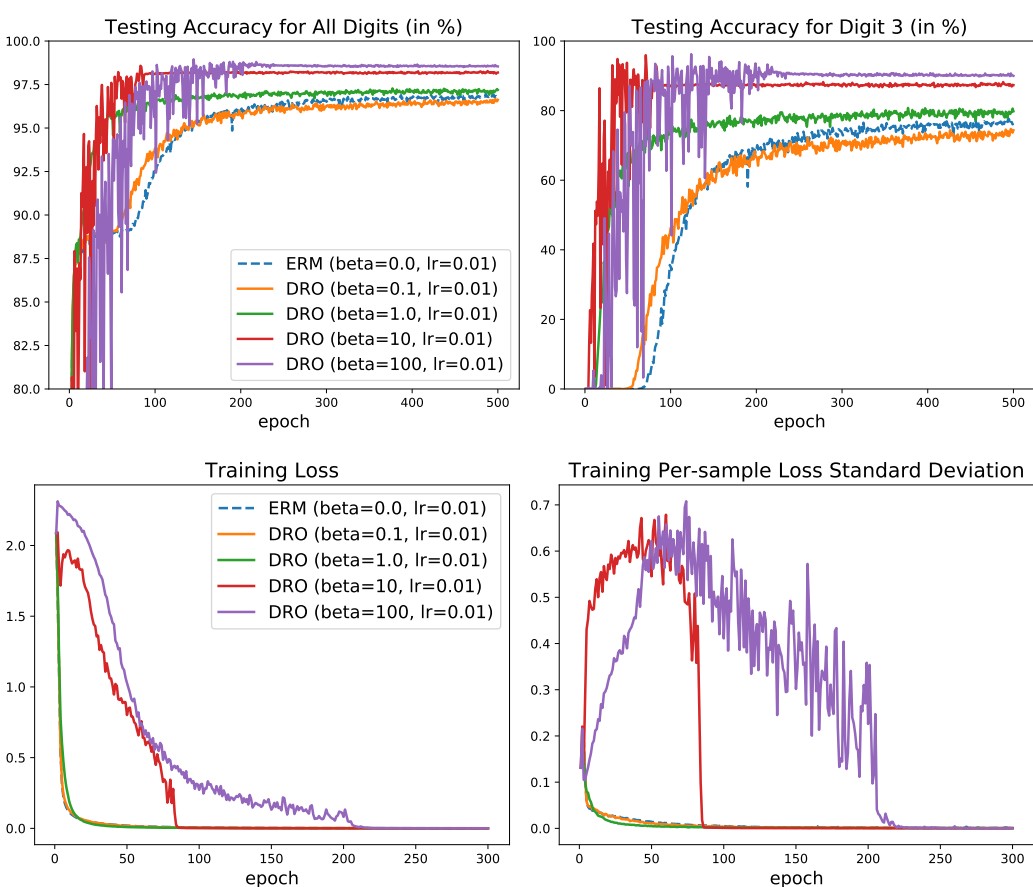

Figure 2: Comparison of learning curves at testing (top panels) and at training (bottom panels) for ERM with SGD (blue) and DRO with our SGD with hardness weighted sampling for different values of $\beta$ on MNIST ($\beta = 0.1$, $\beta = 1$, $\beta = 10$, $\beta = 100$). The models are trained on an imbalanced MNIST dataset (only $1\%$ of the digits 3 kept in the training datatset) and evaluated on the original MNIST testing dataset.

### A.1.1 IMPLEMENTATION DETAILS

For our experiments on MNIST, we used a Wide Residual Network (WRN) (Zagoruyko & Komodakis, 2016). The familly of WRN models has proved to be very efficient and flexible, achieving state-of-the-art accuracy on several dataset. More specifically, we used WRN-16-1 (see Zagoruyko & Komodakis, 2016, section 2.3).

For the optimization we used a learning rate of $0.01$. No momentum or weight decay were used. No data augmentation was used.

### A.1.2 COMMENT ON EARLY-STOPPING WITH ERM AND DRO

It is worth noting that we do not use the knowledge of the training dataset bias in our method. For a less obvious bias, we would have access to the accuracy on the under-represented subset of the training set. As a result, in general, convergence criteria can be based only on the global measure of accuracy (in our case the top left panel of Figure 2).

Early-stopping is a widely used heuristic for selecting the optimal number of epochs to prevent overfitting (Bengio, 2012). Early-stopping crucially depends on the *patience* parameter. In the top left panel of Figure 2, there is no improvement of the accuracy of ERM of more than $0.04\%$ between epoch 20 and 30. As a result, if the *patience* parameter is too low, ERM might be considered to plateau at an epoch at which it ERM achieves an accuracy of $0\%$ on the under-represented class.

This suggests two things. First, the mean accuracy is not a good criteria to decide when to stop the training of ERM for safety-critical systems. Second, our SGD with hardness weighted sampling converge faster than SGD to a safe solution, and is more robust to the hyperparameters of early-stopping.

# B    CONVERGENCE OF SGD WITH HARDNESS WEIGHTED SAMPLING FOR OVER-PARAMETERIZED DEEP NEURAL NETWORKS WITH ReLU (DETAILED STATEMENT)

In this section, we give a detailed statement of the convergence results presented in 4.3.

Our analysis is based on the results developed in (Allen-Zhu et al., 2019) which is a simplified version of (Allen-Zhu et al., 2018). Improving on those theoretical results would automatically improve our results as well. We focus on providing theoretical tools that could be used to generalize any convergence result for ERM using SGD to DRO using Algorithm 4.1. In addition, we compared our conditions and results the one obtained for ERM and SGD in (Allen-Zhu et al., 2019) and discuss the implication of those differences.

Let us first state our assumptions on the neural network $h$, and the per-example loss function $\mathcal{L}$.

**Assumption B.1** (Deep Neural Network). *In this section, we use the following notations and assumptions similar to (Allen-Zhu et al., 2019):*

- *$h$ is a fully connected neural network with $L + 2$ layers, ReLU activation function, and $m$ nodes in each hidden layers*

- *For all $i \in \{1, \dots, n\}$, we denote $h_i : \boldsymbol{\theta} \mapsto h_i(x_i; \boldsymbol{\theta})$ the output $d$ dimensional scores of $h$ applied to example $x_i$ of dimension $\mathfrak{d}$.*

- *$\boldsymbol{\theta} = (\boldsymbol{\theta}_l)_{l=0}^{L+1}$ is the set of parameters of the neural network $h$, where $\boldsymbol{\theta}_l$ is the set of weights for layer $l$ with $\boldsymbol{\theta}_0 \in \mathbb{R}^{\mathfrak{d} \times m}$, $\boldsymbol{\theta}_{L+1} \in \mathbb{R}^{m \times d}$, and $\boldsymbol{\theta}_l \in \mathbb{R}^{m \times m}$ for any other $l$.*

- *(Data separation) It exists $\delta > 0$ such that for all $i, j \in \{1, \dots, n\}$, if $i \neq j$, $\|x_i - x_j\| \geq \delta$.*

- *We assume $m \geq \Omega(d \times \text{poly}(n, L, \delta^{-1}))$ for some sufficiently large polynomial $\text{poly}$, and $\delta \geq O\left(\frac{1}{L}\right)$. We refer the reader to (Allen-Zhu et al., 2019) for details about $\text{poly}$.*

- *The parameters $\boldsymbol{\theta} = (\boldsymbol{\theta}_l)_{l=0}^{L+1}$ are initialized at random such that:*

  - *$[\boldsymbol{\theta}_0]_{i,j} \sim \mathcal{N}\left(0, \frac{2}{m}\right)$ for every $(i, j) \in \{1, \dots, m\} \times \{1, \dots, \mathfrak{d}\}$*
  - *$[\boldsymbol{\theta}_l]_{i,j} \sim \mathcal{N}\left(0, \frac{2}{m}\right)$ for every $(i, j) \in \{1, \dots, m\}^2$ and $l \in \{1, \dots, L\}$*
  - *$[\boldsymbol{\theta}_{L+1}]_{i,j} \sim \mathcal{N}\left(0, \frac{1}{d}\right)$ for every $(i, j) \in \{1, \dots, d\} \times \{1, \dots, m\}$*

**Assumption B.2** (Regularity of $\mathcal{L}$). *For all $i$, $\mathcal{L}_i$ is a $C(\nabla \mathcal{L})$-gradient Lipschitz continuous, $C(\mathcal{L})$-Lipschitz continuous, and bounded (potentially non-convex) function.*

We first generalize the convergence of SGD in (Allen-Zhu et al., 2019, Theorem 2) to the minimization of the distributionally robust loss using SGD and an *exact* hardness weighted sampling (19), i.e. with an exact non-stale loss history.

**Theorem B.1** (Convergence of Robust SGD with exact Loss History). *Let batch size $1 \leq b \leq n$, and $\epsilon > 0$. Suppose there exists constants $C_1, C_2, C_3 > 0$ such that the number of hidden units satisfies $m \geq C_1(d\epsilon^{-1} \times \text{poly}(n, L, \delta^{-1}))$, $\delta \geq \left(\frac{C_2}{L}\right)$, and the learning rate be $\eta_{exact} = C_3\left(\min\left(1, \frac{\alpha n^2 \rho}{\beta C(\mathcal{L})^2 + 2n\rho C(\nabla \mathcal{L})}\right) \times \frac{b\delta d}{\text{poly}(n, L)m \log^2(m)}\right)$. There exists constants $C_4, C_5 > 0$ such*

*that with probability at least $1 - \exp\left(-C_4(\log^2(m))\right)$ over the randomness of the initialization and the mini-batches, Robust SGD with exact loss vector finds $\|\nabla_{\boldsymbol{\theta}}(R \circ \mathcal{L} \circ \boldsymbol{h})(\boldsymbol{\theta})\| \leq \epsilon$ after $T = C_5\left(\frac{Ln^3}{\eta_{exact}\delta\epsilon^2}\right)$ iterations.*

where $\alpha = \min_{\boldsymbol{\theta}} \min_i \bar{p}_i(\mathcal{L}(\boldsymbol{\theta}))$ is lower bound on the sampling probabilities. For the Kullback-Leibler $\phi$-divergence, and for any $\phi$-divergence satisfying assumption 3.1 with a robustness parameter $\beta$ small enough, we have $\alpha > 0$. We refer the reader to (Allen-Zhu et al., 2019, Theorem 2) for the values of the constants $C_1$, $C_2$, $C_3$, $C_4$, $C_5$ and the definitions of the polynomials. Compared to (Allen-Zhu et al., 2019, Theorem 2) only the learning rate differs. The $\min(1, .)$ operation in the formula for $\eta_{exact}$ allows us to guarantee that $\eta_{exact} \leq \eta'$ where $\eta'$ is the learning rate of (Allen-Zhu et al., 2019, Theorem 2). The proof can be found in Appendix D.7.3.

It is worth noting that for the KL $\phi$-divergence, $\rho = \frac{1}{n}$. In addition, in the limit $\beta \to 0$, which corresponds to ERM, we have $\alpha \to \frac{1}{n}$. As a result, we recover exactly Theorem 2 of (Allen-Zhu et al., 2019) as extended in their Appendix A for any $\mathcal{L}$ that satisfies assumption $B.2$ with $C(\nabla \mathcal{L}) = 1$.

When the amount of distributionally robustness increases the sampling differs more and more from the uniform sampling and becomes more sensitive to changes of the loss distribution. One way to mitigate this issue is to reduce the learning rate. The conditions of Theorem B.1 are consistent with this observation since when $\beta$ increases, $\alpha$ and $\eta_{exact}$ decreases.

In practice in algorithm 4.1, we have access only to a stale loss history. We know restate the convergence of Robust SGD with a stale loss history and a warm-up as in Algorithm 4.1.

**Theorem B.2** (Convergence of Robust SGD with Stale Loss History and warm-up). *Let batch size $1 \leq b \leq n$, and $\epsilon > 0$. Under the conditions of Theorem B.1, the same notations, and with the learning rate $\eta_{stale} = C_6 \min\left(1, \frac{\alpha\rho d^{3/2}\delta b \log\left(\frac{1}{1-\alpha}\right)}{\beta C(\mathcal{L})A(\nabla \mathcal{L})Lm^{3/2}n^{3/2}\log^2(m)}\right) \times \eta_{exact}$ for a constant $C_6 > 0$.*

*With probability at least $1 - \exp\left(-C_4(\log^2(m))\right)$ over the randomness of the initialization and the mini-batches, Robust SGD with exact loss vector finds $\|\nabla_{\boldsymbol{\theta}}(R \circ \mathcal{L} \circ \boldsymbol{h})(\boldsymbol{\theta})\| \leq \epsilon$ after $T = C_5\left(\frac{Ln^3}{\eta_{stale}\delta\epsilon^2}\right)$ iterations.*

Where $C(\mathcal{L}) > 0$ is a constant such that $\mathcal{L}$ is $C(\mathcal{L})$-Lipschitz continuous, and $A(\nabla \mathcal{L}) > 0$ is a constant that bound the gradient of $\mathcal{L}$ with respect to its input. $C(\mathcal{L})$ and $A(\nabla \mathcal{L})$ are guaranteed to exist under assumptions B.1. The proof can be found in Appendix D.8.

Compared to Theorem B.1 only the learning rate differs. Similarly to Theorem B.1, when $\beta$ tends to zero we recover Theorem 2 of (Allen-Zhu et al., 2019).

It is worth noting that when $\beta$ increases, $\frac{\alpha\rho d^{3/2}\delta b \log\left(\frac{1}{1-\alpha}\right)}{\beta C(\mathcal{L})A(\nabla \mathcal{L})Lm^{3/2}n^{3/2}\log^2(m)}$ decreases. This implies that $\eta_{stale}$ decreases faster than $\eta_{exact}$ when $\beta$ increases. This was to be expected since the error that is made by using the stale loss history instead of the exact loss increases when $\beta$ increases.

## C    SUMMARY OF THE NOTATIONS USED IN THE PROOFS

For the ease of following the proofs we first summarize our notations.

### C.1    PROBABILITY THEORY NOTATIONS

- $\Delta_n = \{p = (p_i)_{i=1}^n \in [0,1]^n, \ \sum_i p_i = 1\}$
- Let $q = (q_i) \in \Delta_n$, and $f$ a function, we denote $\mathbb{E}_q[f(\mathrm{x})] := \sum_{i=1}^n q_i f(x_i)$.
- Let $q \in \Delta_n$, and $f$ a function, we denote $\mathbb{V}_q[f(\mathrm{x})] := \sum_{i=1}^n q_i \|f(x_i) - \mathbb{E}_q[f(\mathrm{x})]\|^2$.
- $\hat{p}_{\text{data}}$ is the uniform training data distribution, i.e. $\hat{p}_{\text{data}} = \left(\frac{1}{n}\right)_{i=1}^n \in \Delta_n$

### C.2    MACHINE LEARNING NOTATIONS

- n is the number of training examples

- d is the dimension of the output
- $\mathfrak{d}$ is the dimension of the input
- training data: $\{(x_i, y_i)\}_{i=1}^n$, where for all $i \in \{1, \ldots, n\}$, $x_i \in \mathbb{R}^{\mathfrak{d}}$ and $y_i \in \mathbb{R}^d$
- $h : \boldsymbol{x} \mapsto \boldsymbol{y}$ is the predictor
- $\boldsymbol{\theta}$ is the set of parameters of the predictor
- For all $i$, $h_i : \boldsymbol{\theta} \mapsto h(\mathrm{x}_i; \boldsymbol{\theta})$ is the output of the network for example $i$ as a function of $\boldsymbol{\theta}$
- $\mathcal{L}$ is the objective function
- $\mathcal{L}_i : h_i \mapsto \mathcal{L}(h_i, y_i)$ is the objective function for example $i$.
- By abuse of notation we also denote by $\mathcal{L}$ the function $\mathcal{L} : (h_i)_{i=1}^n \mapsto (\mathcal{L}_i(h_i))_{i=1}^n$
- $b \in \{1, \ldots, n\}$ is the batch size
- $\eta > 0$ is the learning rate
- ERM is short for Empirical Risk Minimization

## C.3 DISTRIBUTIONALLY ROBUST OPTIMISATION NOTATIONS

- Forall $\boldsymbol{\theta}$, $R(\mathcal{L}(\boldsymbol{h}(\boldsymbol{\theta}))) = \max_{q \in \Delta_n} \mathbb{E}_q \left[ \mathcal{L}\left(h(\mathrm{x}; \boldsymbol{\theta}), \boldsymbol{y}\right) \right] - \frac{1}{\beta} D_\phi(q \| \hat{p}_{\mathrm{data}})$ is the **Distributionally Robust Loss** evaluated at $\boldsymbol{\theta}$, where $\beta > 0$ is the parameter that adjusts the distributionally robustness (see (9) for more details). For short, we also used the term **Robust Loss** for $R(\mathcal{L}(\boldsymbol{h}(\boldsymbol{\theta})))$
- DRO is short for Distributionally Robust Optimisation

## C.4 MISCELLANEOUS

By abuse of notation, and similarly to (Allen-Zhu et al., 2019), we use the Bachmann-Landau notations to hide constants that do not depend on our main hyper-parameters. Let $f$ and $g$ be two scalars, we note:

$$\begin{cases} f \leq O(g) & \iff & \exists c > 0 & \text{s.t.} & f \leq cg \\ f \geq \Omega(g) & \iff & \exists c > 0 & \text{s.t.} & f \geq cg \\ f = \Theta(g) & \iff & \exists c_1 > 0 \text{ and } \exists c_2 > c_1 & \text{s.t.} & c_1 g \leq f \leq c_2 g \end{cases}$$

# D PROOFS

## D.1 ASSUMPTIONS FOR THE $\phi$-DIVERGENCE

Let $D_\phi$ be a $\phi$-Divergence, and $c \in \mathbb{R}$. One can note that for $\phi_c : z \mapsto \phi(z) + c(1 - z)$, we have for all $p = (p_i)_{i=1}^n, q = (q_i)_{i=1}^n \in \Delta_n$,

$$\begin{aligned} D_{\phi_c}(q \| p) &= \sum_{i=1}^n p_i \phi_c\left(\frac{q_i}{p_i}\right) \\ &= \sum_{i=1}^n p_i \left( \phi\left(\frac{q_i}{p_i}\right) + c\left(1 - \frac{q_i}{p_i}\right) \right) \\ &= D_\phi(q \| p) + c\left( \sum_{i=1}^n p_i - \sum_{i=1}^n q_i \right) \\ &= D_\phi(q \| p) \end{aligned} \tag{23}$$

In other words, $D_\phi$ is not uniquely defined by $\phi$ under the definition 3.3. If $\phi$ is differentiable, one can assume without loss of generality that $\phi'(1) = 0$.

For more detailed discussion, we refer the interested reader to (Csiszár et al., 2004).

### D.2 Proof of Example 4.1: formula of the sampling probabilities for the KL divergence

We give here a simple proof of the formula of the sampling probabilities for the KL divergence as $\phi$-divergence (i.e. $\phi : z \mapsto z \log(z) - z + 1$)

$$\forall \boldsymbol{\theta}, \quad \bar{p}(\mathcal{L}(\boldsymbol{h}(\boldsymbol{\theta}))) = \text{softmax}\left(\beta \, \mathcal{L}(\boldsymbol{h}(\boldsymbol{\theta}))\right)$$

For any $\boldsymbol{\theta}$, the distributionally robust loss (9) for the KL divergence at $\boldsymbol{\theta}$ is given by

$$R \circ \mathcal{L} \circ \boldsymbol{h}(\boldsymbol{\theta}) = \max_{q \in \Delta_n} \left( \sum_{i=1}^n q_i \, \mathcal{L}_i \circ h_i(\boldsymbol{\theta}) - \frac{1}{\beta} \sum_{i=1}^n q_i \log\left(nq_i\right) \right)$$

$$= \max_{q \in \Delta_n} \sum_{i=1}^n \left( q_i \, \mathcal{L}_i \circ h_i(\boldsymbol{\theta}) - \frac{1}{\beta} q_i \log\left(nq_i\right) \right)$$

To simplify the notations, let us denote $\boldsymbol{v} = (v_i)_{i=1}^n = \mathcal{L} \circ \boldsymbol{h}(\boldsymbol{\theta}) = (\mathcal{L}_i \circ h_i(\boldsymbol{\theta}))_{i=1}^n$, and $\bar{p} = (\bar{p}_i)_{i=1}^n = \bar{p}(\mathcal{L}(\boldsymbol{h}(\boldsymbol{\theta})))$.

Thus $\bar{p}(\mathcal{L}(\boldsymbol{h}(\boldsymbol{\theta})))$ is, by definition, solution of the optimization problem

$$\arg\max_{q \in \Delta_n} \sum_{i=1}^n \left( q_i v_i - \frac{1}{\beta} q_i \log\left(nq_i\right) \right) \tag{24}$$

First, let us remark that the function $q \mapsto \sum_{i=1}^n q_i \log\left(nq_i\right)$ is strictly convex on the non empty closed convex set $\Delta_n$ as a sum of strictly convex functions. This implies that the optimization (24) has a unique solution and as a result $\bar{p}(\mathcal{L}(\boldsymbol{h}(\boldsymbol{\theta})))$ is well defined.

We now reformulate the optimization problem (24) as a convex smooth constrained optimization problem by writing the condition $q \in \Delta_n$ as constraints.

$$\arg\max_{q \in \mathbb{R}_+^n} \sum_{i=1}^n \left( q_i v_i - \frac{1}{\beta} q_i \log\left(nq_i\right) \right)$$
$$\text{s.t.} \sum_{i=1}^n q_i = 1 \tag{25}$$

There exists a Lagrange multiplier $\lambda \in \mathbb{R}$, such that the solution $\bar{p}$ of (25) is characterized by

$$\forall i \in \{1, \ldots, n\}, \quad v_i - \frac{1}{\beta}\left(\log\left(n\bar{p}_i\right) + 1\right) + \lambda = 0$$
$$\sum_{i=1}^n \bar{p}_i = 1 \tag{26}$$

Which we can rewrite as

$$\forall i \in \{1, \ldots, n\}, \quad \bar{p}_i = \frac{1}{n} \exp\left(\beta\left(v_i + \lambda\right) - 1\right)$$
$$\frac{1}{n} \sum_{i=1}^n \exp\left(\beta\left(v_i + \lambda\right) - 1\right) = 1 \tag{27}$$

The last equality gives

$$\exp\left(\beta\lambda - 1\right) = \frac{n}{\sum_{i=1}^n \exp\left(\beta v_i\right)}$$

And by replacing in the formula of the $\bar{p}_i$

$$\forall i \in \{1, \ldots, n\}, \quad \bar{p}_i = \frac{1}{n} \exp\left(\beta v_i\right) \exp\left(\beta\lambda - 1\right)$$
$$= \frac{\exp\left(\beta v_i\right)}{\sum_{j=1}^n \exp\left(\beta v_j\right)}$$

Which corresponds exactly to

$$\bar{p} = \text{softmax}\left(\beta \boldsymbol{v}\right)$$

### D.3 PROOF OF LEMMA 4.1: REGULARITY PROPERTIES OF $R$

For the ease of reading, let us first recall that given a $\phi$-Divergence that satisfies assumptions 3.1, we have defined in (9)

$$R : \mathbb{R}^n \to \mathbb{R}$$
$$v \mapsto \max_{q \in \Delta_n} \sum_i q_i v_i - \frac{1}{\beta} D_\phi(q \| p_{train}) \tag{28}$$

And in (13)

$$G : \mathbb{R}^n \to \mathbb{R}$$
$$p \mapsto \frac{1}{\beta} D_\phi(p \| p_{train}) + \delta_{\Delta_n}(p) \tag{29}$$

where $\delta_{\Delta_n}$ is the characteristic function of the closed convex set $\Delta_n$, i.e.

$$\forall p \in \mathbb{R}^n, \ \delta_{\Delta_n}(p) = \begin{cases} 0 & \text{if } p \in \Delta_n \\ +\infty & \text{otherwise} \end{cases} \tag{30}$$

We now prove Lemma 4.1 on the regularity of R.

**Lemma D.1** (Regularity of R – Restated from Lemma 4.1). *Let $\phi$ that satisfies Assumption 3.1, $G$ and $R$ satisfy*

$$G \text{ is } \left( \frac{n\rho}{\beta} \right) \text{-strongly convex} \tag{31}$$

$$R(\mathcal{L}(h(\boldsymbol{\theta}))) = \max_{q \in \mathbb{R}^n} \left( \langle \mathcal{L}(\boldsymbol{h}(\boldsymbol{\theta})), q \rangle - G(q) \right) = G^* \left( \mathcal{L}(\boldsymbol{h}(\boldsymbol{\theta})) \right) \tag{32}$$

$$R \text{ is } \left( \frac{\beta}{n\rho} \right) \text{-gradient Lipschitz continuous.} \tag{33}$$

$\phi$ is $\rho$-strongly convex on $[0, n]$ so

$$\forall x, y \in [0, n]^2, \forall \lambda \in [0, 1], \phi(\lambda x + (1 - \lambda)y) \leq \lambda \phi(x) + (1 - \lambda)\phi(y) - \frac{\rho\lambda(1 - \lambda)}{2}|y - x|^2 \tag{34}$$

Let $p = (p_i)_{i=1}^n$, $q = (q_i)_{i=1}^n \in \Delta_n$, and $\lambda \in [0, 1]$, using (34) and the convexity of $\delta_{\Delta_n}$, we obtain:

$$G(\lambda p + (1 - \lambda)q) = \frac{1}{\beta n} \sum_{i=1}^n \phi(n\lambda p_i + n(1 - \lambda)q_i) + \delta_{\Delta_n}(\lambda p + (1 - \lambda)q)$$

$$\leq \lambda G(p) + (1 - \lambda)G(q) - \frac{1}{\beta n} \sum_{i=1}^n \frac{\rho\lambda(1 - \lambda)}{2}|nq_i - np_i|^2 \tag{35}$$

$$\leq \lambda G(p) + (1 - \lambda)G(q) - \frac{n\rho}{\beta}\frac{\lambda(1 - \lambda)}{2} \|q - p\|^2$$

This proves that $G$ is $\frac{n\rho}{\beta}$-strongly convex.

Since $G$ is convex, $R = G^*$ is also convex, and $R^* = (G^*)^* = G$ (Hiriart-Urruty & Lemaréchal, 2013).

We obtain (32) using Definition 4.1.

We now show that $R$ is Frechet differentiable on $\mathbb{R}^n$. Let $v \in \mathbb{R}^n$.

$G$ is strongly-convex, so in particular $G$ is strictly convex. This implies that the following optimization problem has a unique solution that we denote $\hat{p}(v)$.

$$\arg\max_{q \in \mathbb{R}^n} \left( \langle v, q \rangle - G(q) \right) \tag{36}$$

In addition

$$\hat{p} \in \Delta_n \text{ solution of (36)} \iff 0 \in v - \partial G(\hat{p})$$
$$\iff v \in \partial G(\hat{p})$$
$$\iff \hat{p} \in \partial G^*(v)$$
$$\iff \hat{p} \in \partial R(v)$$

where we have used (Hiriart-Urruty & Lemaréchal, 2013, Proposition 6.1.2 p.39) for the third equivalence, and (32) for the last equivalence.

As a result, $\partial R(v) = \{\hat{p}(v)\}$. this implies that $R$ admit a gradient at $v$, and

$$\nabla_v R(v) = \hat{p}(v) \tag{37}$$

Since this holds for any $v \in \mathbb{R}^n$, we deduce that $R$ is Frechet differentiable on $\mathbb{R}^n$.

We are now ready to show that $R$ is $\frac{\beta}{n\rho}$-gradient Lipchitz continuous by using the following lemma (Hiriart-Urruty & Lemaréchal, 2013, Theorem 6.1.2 p.280).

**Lemma D.2.** *A necessary and sufficient condition for a convex function $f : \mathbb{R}^n \to \mathbb{R}$ to be c-strongly convex on a convex set $C$ is that for all $x_1, x_2 \in C$*

$$\langle s_2 - s_1, x_2 - x_1 \rangle \geq c \|x_2 - x_1\|^2 \quad \text{for all } s_i \in \partial f(x_i), i = 1, 2.$$

Using this lemma for $f = G$, $c = \frac{n\rho}{\beta}$, and $C = \Delta_n$, we obtain:

For all $p_1, p_2 \in \Delta_n$, for all $v_1 \in \partial G(p_1)$, $v_2 \in \partial G(p_2)$,

$$\langle v_2 - v_1, p_2 - p_1 \rangle \geq \frac{n\rho}{\beta} \|p_2 - p_1\|^2$$

In addition, for $i \in \{1, 2\}$, $v_i \in \partial G(p_i) \iff p_i \in \partial R(v_1) = \{\nabla_v R(v_i)\}$.

And using Cauchy Schwarz inequality

$$\|v_2 - v_1\| \|p_2 - p_1\| \geq \langle v_2 - v_1, p_2 - p_1 \rangle$$

We conclude that

$$\frac{n\rho}{\beta} \|\nabla_v R(v_2) - \nabla_v R(v_1)\| \leq \|v_2 - v_1\|$$

Which implies that $R$ is $\frac{\beta}{n\rho}$-gradient Lipchitz continuous.

### D.4 Proof of Lemma 4.2: Formula of the distributionally robust loss gradient

We prove Lemma 4.2 that we restate here for the ease of reading.

**Lemma D.3** (Stochastic Gradient of the Distributionally Robust Loss – Restated from Lemma 4.2)**.** *For all $\boldsymbol{\theta}$, we have*

$$\bar{p}(\mathcal{L}(\boldsymbol{h}(\boldsymbol{\theta}))) = \nabla_{\boldsymbol{v}} R(\mathcal{L}(\boldsymbol{h}(\boldsymbol{\theta}))) \tag{38}$$

$$\nabla_{\boldsymbol{\theta}}(R \circ \mathcal{L} \circ \boldsymbol{h})(\boldsymbol{\theta}) = \mathbb{E}_{\bar{p}(\mathcal{L}(\boldsymbol{h}(\boldsymbol{\theta})))} \left[ \nabla_{\boldsymbol{\theta}} \mathcal{L}(h(\mathrm{x}; \boldsymbol{\theta}), y) \right] \tag{39}$$

where $\nabla_{\boldsymbol{v}} R$ is the gradient of $R$ with respect to its input.

For a given $\boldsymbol{\theta}$, equality (38) is a special case of (37) for $\boldsymbol{v} = \mathcal{L}(\boldsymbol{h}(\boldsymbol{\theta}))$.

Then using the chain rule and (38),

$$\nabla_{\boldsymbol{\theta}}(R \circ \mathcal{L} \circ \boldsymbol{h})(\boldsymbol{\theta}) = \sum_{i=1}^n \frac{\partial R}{\partial v_i}(\mathcal{L} \circ \boldsymbol{h}(\boldsymbol{\theta})) \nabla_{\boldsymbol{\theta}}(\underset{i}{\mathcal{L}} \circ h_i)(\boldsymbol{\theta})$$

$$= \sum_{i=1}^n \bar{p}_i(\mathcal{L}(\boldsymbol{h}(\boldsymbol{\theta}))) \nabla_{\boldsymbol{\theta}}(\underset{i}{\mathcal{L}} \circ h_i)(\boldsymbol{\theta})$$

$$= \mathbb{E}_{\bar{p}(\mathcal{L}(\boldsymbol{h}(\boldsymbol{\theta})))} \left[ \nabla_{\boldsymbol{\theta}} \mathcal{L}(h(\mathrm{x}; \boldsymbol{\theta}), y) \right]$$

### D.5 Proof of Theorem 4.2: Distributionally Robust Optimization as Principled Hard Example Mining

Let $D_\phi$ an $\phi$-divergence satisfying Assumption 3.1, and $v = (v_i)_{i=1}^n \in \mathbb{R}^n$. $v$ will play the role of a generic loss vector.

$\phi$ is strongly convex, and $\Delta^n$ is closed and convex, so the following optimization problem has one and only one solution:

$$\max_{p=(p_i)_{i=1}^n \in \Delta^n} \langle v, p \rangle - \frac{1}{\beta n} \sum_{i=1}^n \phi(np_i) \tag{40}$$

Making the constraints associated with $p \in \Delta_n$ explicit, this can be rewritten as

$$\max_{p=(p_i)_{i=1}^n \in \mathbb{R}^n} \langle v, p \rangle - \frac{1}{\beta n} \sum_{i=1}^n \phi(np_i)$$
$$\text{s.t. } \forall i \in \{1, \ldots, n\}, \ p_i \geq 0 \tag{41}$$
$$\text{s.t. } \sum_{i=1}^n p_i = 1$$

There exists KKT multipliers $\lambda \in \mathbb{R}$ and $\forall i, \mu_i \geq 0$ such that the solution $\bar{p} = (\bar{p}_i)_{i=1}^n$ satisfies:

$$\begin{cases} \forall i \in \{1, \ldots, n\}, & v_i - \frac{1}{\beta} \phi'(n\bar{p}_i) + \lambda - \mu_i = 0 \\ \forall i \in \{1, \ldots, n\}, & \mu_i p_i = 0 \\ \forall i \in \{1, \ldots, n\}, & p_i \geq 0 \\ & \sum_{i=1}^n \bar{p}_i = 1 \end{cases} \tag{42}$$

Since $\phi$ is continuously differentiable and strongly convex, we have $(\phi')^{-1} = (\phi^*)'$, where $\phi^*$ is the Fenchel conjugate of $\phi$ (see Hiriart-Urruty & Lemaréchal, 2013, Proposition 6.1.2). As a result, (42) can be rewritten has:

$$\begin{cases} \forall i \in \{1, \ldots, n\}, & \bar{p}_i = \frac{1}{n} (\phi^*)' (\beta(v_i + \lambda - \mu_i)) \\ \forall i \in \{1, \ldots, n\}, & \mu_i p_i = 0 \\ \forall i \in \{1, \ldots, n\}, & p_i \geq 0 \\ & \frac{1}{n} \sum_{i=1}^n (\phi^*)' (\beta(v_i + \lambda - \mu_i)) = 1 \end{cases} \tag{43}$$

We now show that the KKT multipliers are uniquely defined.

**The $\mu_i$'s are uniquely defined by $v$ and $\lambda$:**
Since $\forall i \in \{1, \ldots, n\}$, $\mu_i p_i = 0$, $p_i \geq 0$ and $\mu_i \geq 0$, for all $\forall i \in \{1, \ldots, n\}$, either $p_i = 0$ or $\mu_i = 0$.

In the case $p_i = 0$, using (43) it comes $(\phi^*)' (\beta(v_i + \lambda - \mu_i)) = 0$.

According to assumption 3.1, $\phi$ is strongly convex and continuously differentiable, so $\phi'$ and $(\phi^*)' = (\phi')^{-1}$ are continuous and strictly increasing functions. As a result, it exists a unique $\mu_i$ (dependent to $v$ and $\lambda$) such that:

$$(\phi^*)' (\beta(v_i + \lambda - \mu_i)) = 0$$

And (43) can be rewritten as:

$$\begin{cases} \forall i \in \{1, \ldots, n\}, & \bar{p}_i = \text{ReLU} \left( \frac{1}{n} (\phi^*)' (\beta(v_i + \lambda)) \right) = \frac{1}{n} \text{ReLU} \left( (\phi^*)' (\beta(v_i + \lambda)) \right) \\ & \frac{1}{n} \sum_{i=1}^n \text{ReLU} \left( (\phi^*)' (\beta(v_i + \lambda)) \right) = 1 \end{cases} \tag{44}$$

**$\lambda$ is uniquely defined by $v$ and a continuous function of $v$:**
Let $\lambda \in \mathbb{R}$ that satisfies (44).

We have $\frac{1}{n} \sum_{i=1}^{n} \text{ReLU} \left( (\phi^*)' \left( \beta(v_i + \lambda) \right) \right) = 1$. So there exists at least one index $i_0$ such that

$$\text{ReLU} \left( (\phi^*)' \left( \beta(v_{i_0} + \lambda) \right) \right) = (\phi^*)' \left( \beta(v_{i_0} + \lambda) \right) \geq 1$$

Since $(\phi^*)^{-1}$ is continuous and striclty increasing, $\lambda' \mapsto \text{ReLU} \left( (\phi^*)' \left( \beta(v_{i_0} + \lambda') \right) \right)$ is continuous and strictly increasing on a neighborhood of $\lambda$.

In addition ReLU is continuous and increasing, so for all $i \in \{1, \ldots, n\}$, $\lambda' \mapsto \text{ReLU} \left( (\phi^*)' \left( \beta(v_i + \lambda') \right) \right)$ is a continuous and increasing function.

As a result, $\lambda' \mapsto \frac{1}{n} \sum_{i=1}^{n} \text{ReLU} \left( (\phi^*)' \left( \beta(v_i + \lambda') \right) \right)$ is a continuous function that is increasing on $\mathbb{R}$, and strictly increasing on a neighborhood of $\lambda$.

This implies that $\lambda$ is uniquely defined by $v$, and that $v \mapsto \lambda(v)$ is continuous.

**Hard Example Mining Sampling:**

For any pseudo loss vector $v = (v_i)_{i=1}^{n} \in \mathbb{R}^n$, there exists a unique $\lambda$ and a unique $\bar{p}$ that satisfies (44), so we can define the mapping:

$$\begin{aligned} \bar{p}: \ \mathbb{R}^n &\to \Delta_n \\ v &\mapsto \bar{p}(v; \lambda(v)) \end{aligned} \tag{45}$$

where for all $v$, $\lambda(v)$ is the unique $\lambda \in \mathbb{R}$ satisfying (44).

We will now demonstrate that each $\bar{p}_{i_0}(v)$ for $i_0 \in \{1, \ldots, n\}$ is an increasing function of $v_i$ and a decreasing function of the $v_i$ for $i \neq i_0$. Without loss of generality we assume $i_0 = 1$.

Let $v = (v_i)_{i=1}^{n} \in \mathbb{R}^n$, and $\epsilon > 0$.

Let us define $v' = (v_i')_{i=1}^{n} \in \mathbb{R}^n$, such that $v_1' = v_1 + \epsilon$ and $\forall i \in \{2, \ldots, n\}, \ v_i' = v_i$.

Similarly as in the proof of the uniqueness of $\lambda$ above, we can show that there exists $\eta > 0$ such that the function

$$F : \lambda' \mapsto \frac{1}{n} \sum_{i=1}^{n} \text{ReLU} \left( (\phi^*)' \left( \beta(v_i + \lambda') \right) \right)$$

is continuous and strictly increasing on $[\lambda(v) - \eta, \lambda(v) + \eta]$, and $F(\lambda(v)) = 1$.

$v \mapsto \lambda(v)$ is continuous, so for $\epsilon$ small enough $\lambda(v') \in [\lambda(v) - \eta, \lambda(v) + \eta]$.

Let us now prove by contradiction that $\lambda(v') \leq \lambda(v)$. Therefore, let us assume that $\lambda(v') > \lambda(v)$. Then, as $\text{ReLU} \circ (\phi^*)'$ is an increasing function and $F$ is strictly increasing on $[\lambda(v) - \eta, \lambda(v) + \eta]$, and $\epsilon > 0$ we obtain

$$\begin{aligned} 1 &= \frac{1}{n} \sum_{i=1}^{n} \text{ReLU} \left( (\phi^*)' \left( \beta(v_i' + \lambda(v')) \right) \right) \\ &\geq \frac{1}{n} \sum_{i=1}^{n} \text{ReLU} \left( (\phi^*)' \left( \beta(v_i + \lambda(v')) \right) \right) \\ &\geq F(\lambda(v')) \\ &> F(\lambda(v)) \\ &> 1 \end{aligned}$$

which is a contradiction. As a result

$$\lambda(v') \leq \lambda(v) \tag{46}$$

Using (46), (44), and the fact that $\text{ReLU} \circ (\phi^*)'$ is an increasing function, we obtain for all $i \in \{2, \ldots, n\}$

$$
\begin{aligned}
\bar{p}_i(v') &= \frac{1}{n} \text{ReLU} \left( (\phi^*)' \left( \beta(v'_i + \lambda(v')) \right) \right) \\
&= \frac{1}{n} \text{ReLU} \left( (\phi^*)' \left( \beta(v_i + \lambda(v')) \right) \right) \\
&\leq \frac{1}{n} \text{ReLU} \left( (\phi^*)' \left( \beta(v_i + \lambda(v)) \right) \right) \\
&\leq \bar{p}_i(v)
\end{aligned}
\tag{47}
$$

In addition

$$
\sum_{i=1}^{n} \bar{p}_i(v') = 1 = \sum_{i=1}^{n} \bar{p}_i(v)
$$

So necessarily

$$
\bar{p}_1(v') \geq \bar{p}_1(v)
\tag{48}
$$

This holds for any $i_0$ and any $v$, which concludes the proof.

### D.6  PROOF THAT $R \circ \mathcal{L}$ IS ONE-SIDED GRADIENT LIPCHITZ

This property that $R \circ \mathcal{L}$ is one-sided gradient Lipschitz is a key element for the proof of the semismoothness theorem for the distributionally robust loss Theorem D.1.

Under assumption 3.1, we have shown that $R^*$ is $\frac{\beta}{n\rho}$-gradient Lipchitz continuous. And under assumption B.2, for all $i$, $\mathcal{L}_i$ is $C(\mathcal{L})$-Lipschitz continuous and $C(\nabla \mathcal{L})$-gradient Lipschitz continuous.

Let $z = (z_i)_{i=1}^n, z' = (z'_i)_{i=1}^n \in \mathbb{R}^{dn}$.

We want to show that $R \circ \mathcal{L}$ is one-sided gradient Lipschitz, i.e. we want to prove the existence of a constant $C > 0$, independent to $z$ and $z'$, such that:

$$
\langle \nabla_z (R \circ \mathcal{L})(z) - \nabla_z (R \circ \mathcal{L})(z'), z - z' \rangle \leq C \left\| z - z' \right\|^2
$$

We have

$$
\begin{aligned}
\langle \nabla_z (R \circ \mathcal{L})(z) &- \nabla_z (R \circ \mathcal{L})(z'), z - z' \rangle \\
&= \sum_{i=1}^{n} \langle \nabla_{z_i} (R \circ \mathcal{L})(z) - \nabla_{z_i} (R \circ \mathcal{L})(z'), z_i - z'_i \rangle \\
&= \sum_{i=1}^{n} \langle \bar{p}_i(\mathcal{L}(z)) \nabla_{z_i} \mathcal{L}_i(z_i) - \bar{p}_i(\mathcal{L}(z')) \nabla_{z_i} \mathcal{L}_i(z'_i), z_i - z'_i \rangle \\
&= \sum_{i=1}^{n} \bar{p}_i(\mathcal{L}(z)) \langle \nabla_{z_i} \mathcal{L}_i(z_i) - \nabla_{z_i} \mathcal{L}_i(z'_i), z_i - z'_i \rangle \\
&\quad + \sum_{i=1}^{n} \left( \bar{p}_i(\mathcal{L}(z)) - \bar{p}_i(\mathcal{L}(z')) \right) \langle \nabla_{z_i} \mathcal{L}_i(z'_i), z_i - z'_i \rangle
\end{aligned}
\tag{49}
$$

Where for all $i \in \{1, \ldots, n\}$ we have used the chain rule

$$
\nabla_{z_i} (R \circ \mathcal{L})(z) = \sum_{j=1}^{n} \frac{\partial R^*}{\partial v_j} (\mathcal{L}(z)) \nabla_{z_i} \mathcal{L}_j(z_j) = \bar{p}_i(\mathcal{L}(z)) \nabla_{z_i} \mathcal{L}_i(z_i)
$$

Let

$$
A = \left| \sum_{i=1}^{n} \bar{p}_i(\mathcal{L}(z)) \langle \nabla_{z_i} \mathcal{L}_i(z_i) - \nabla_{z_i} \mathcal{L}_i(z'_i), z_i - z'_i \rangle \right|
$$

For all $i$, $\mathcal{L}_i$ is $C(\nabla \mathcal{L})$-gradient Lipchitz continuous, so using Cauchy-Schwarz inequality

$$A \leq \sum_{i=1}^{n} C(\nabla \mathcal{L}) \|z_i - z_i'\|^2 = C(\nabla \mathcal{L}) \|z - z'\|^2 \tag{50}$$

Let

$$B = \left| \sum_{i=1}^{n} (\bar{p}_i(\mathcal{L}(z)) - \bar{p}_i(\mathcal{L}(z'))) \langle \nabla_{z_i} \mathcal{L}_i(z_i'), z_i - z_i' \rangle \right|$$

Using the triangular inequality:

$$\begin{aligned}
B &\leq \left| \sum_{i=1}^{n} (\bar{p}_i(\mathcal{L}(z)) - \bar{p}_i(\mathcal{L}(z'))) (\mathcal{L}_i(z_i) - \mathcal{L}_i(z_i')) \right| \\
&\quad + \left| \sum_{i=1}^{n} (\bar{p}_i(\mathcal{L}(z)) - \bar{p}_i(\mathcal{L}(z'))) (\mathcal{L}_i(z_i') + \langle \nabla_{z_i} \mathcal{L}_i(z_i'), z_i - z_i' \rangle - \mathcal{L}_i(z_i)) \right| \\
&\leq \langle \nabla(R^*)(\mathcal{L}(z)) - \nabla(R^*)(\mathcal{L}(z')), \mathcal{L}(z) - \mathcal{L}(z') \rangle \\
&\quad + 2 \sum_{i=1}^{n} \left| \mathcal{L}_i(z_i') + \langle \nabla_{z_i} \mathcal{L}_i(z_i'), z_i - z_i' \rangle - \mathcal{L}_i(z_i) \right| \\
&\leq \frac{\beta}{n\rho} \|\mathcal{L}(z) - \mathcal{L}(z')\|^2 + 2 \frac{C(\nabla \mathcal{L})}{2} \|z - z'\|^2 \\
&\leq \left( \frac{\beta C(\mathcal{L})^2}{n\rho} + C(\nabla \mathcal{L}) \right) \|z - z'\|^2
\end{aligned} \tag{51}$$

Combining (49), (50) and (51) we finally obtain:

$$\langle \nabla_z (R \circ \mathcal{L})(z) - \nabla_z (R \circ \mathcal{L})(z'), z - z' \rangle \leq \left( \frac{\beta C(\mathcal{L})^2}{n\rho} + 2C(\nabla \mathcal{L}) \right) \|z - z'\|^2 \tag{52}$$

From there, we can obtain the following inequality that will be used for the proof of the semi-smoothness property in Theorem D.1:

$$\begin{aligned}
&R(\mathcal{L}(z')) - R(\mathcal{L}(z)) - \langle \nabla_z (R \circ \mathcal{L})(z), z' - z \rangle \\
&= \int_{t=0}^{1} \langle \nabla_z (R \circ \mathcal{L}) (z + t(z' - z)) - \nabla_z (R \circ \mathcal{L})(z), z' - z \rangle dt \\
&\leq \frac{1}{2} \left( \frac{\beta C(\mathcal{L})^2}{n\rho} + 2C(\nabla \mathcal{L}) \right) \|z - z'\|^2
\end{aligned} \tag{53}$$

### D.7 PROOF OF THE CONVERGENCE OF ROBUST SGD

In this part, we prove the results of Therem B.1 and B.2.

They are generalizations of the convergence result for SGD presented in Theorem 2 of (Allen-Zhu et al., 2019).

For the ease of reading the proof, we remind here the chain rules for the distributionally robust loss (9) that we are going to use intensively in the following proofs.

**Chain rule for the derivative of $R \circ \mathcal{L}$ with respect to the network outputs $h$:**

$$\nabla_h (R \circ \mathcal{L})(h(\boldsymbol{\theta})) = (\nabla_{h_i} (R \circ \mathcal{L})(h(\boldsymbol{\theta})))_{i=1}^{n}$$

$$\forall i \in \{1, \ldots n\}, \quad \nabla_{h_i} (R \circ \mathcal{L})(h(\boldsymbol{\theta})) = \sum_{j=1}^{n} \frac{\partial R}{\partial v_j}(\mathcal{L}(h(\boldsymbol{\theta}))) \nabla_{h_i} \mathcal{L}_j(h_j(\boldsymbol{\theta})) \tag{54}$$

$$= \bar{p}_i(\mathcal{L}(h(\boldsymbol{\theta}))) \nabla_{h_i} \mathcal{L}_i(h_i(\boldsymbol{\theta}))$$

**Chain rule for the derivative of $R \circ \mathcal{L} \circ h$ with respect to the network parameters $\theta$:**

$$
\begin{aligned}
\nabla_{\boldsymbol{\theta}}(R \circ \mathcal{L} \circ h)(\boldsymbol{\theta}) &= \sum_{i=1}^{n} \nabla_{\theta} h_i(\boldsymbol{\theta}) \nabla_{h_i}(R \circ \mathcal{L})(h(\boldsymbol{\theta})) \\
&= \sum_{i=1}^{n} \bar{p}_i(\mathcal{L}(h(\boldsymbol{\theta}))) \nabla_{\theta} h_i(\boldsymbol{\theta}) \nabla_{h_i} \underset{i}{\mathcal{L}}(h_i(\boldsymbol{\theta})) \\
&= \sum_{i=1}^{n} \bar{p}_i(\mathcal{L}(h(\boldsymbol{\theta})) \nabla_{\boldsymbol{\theta}}(\underset{i}{\mathcal{L}} \circ h_i)(\boldsymbol{\theta}))
\end{aligned}
\tag{55}
$$

where for all $i \in \{1, \ldots n\}$, $\nabla_{\theta} h_i(\boldsymbol{\theta})$ is the transpose of the Jacobian matrix of $h_i$ as a function of $\boldsymbol{\theta}$.

### D.7.1 SEMI-SMOOTHNESS PROPERTY FOR THE DISTRIBUTIONALLY ROBUST LOSS

We prove the following lemma which is a generalization of Theorem 4 in (Allen-Zhu et al., 2019) for the distributionally robust loss (9).

**Theorem D.1** (Semi-smoothness of the Distributionally Robust Loss)**.**
*Let $\omega \in \left[\Omega\left(\frac{d^{3/2}}{m^{3/2}L^{3/2}\log^{3/2}(m)}\right), O\left(\frac{1}{L^{4.5}\log^3(m)}\right)\right]$, and the $\boldsymbol{\theta}^{(0)}$ being initialized randomly as described in assumption B.1. With probability as least $1 - \exp\left(-\Omega(m\omega^{3/2}L)\right)$ over the initialization, we have for all $\boldsymbol{\theta}, \boldsymbol{\theta}' \in (\mathbb{R}^{m \times m})^L$ with $\|\boldsymbol{\theta} - \boldsymbol{\theta}^{(0)}\|_2 \leq \omega$, and $\|\boldsymbol{\theta} - \boldsymbol{\theta}'\|_2 \leq \omega$*

$$
\begin{aligned}
R(\mathcal{L}(h(\boldsymbol{\theta}'))) \leq\ & R(\mathcal{L}(h(\boldsymbol{\theta})) + \langle \nabla_{\boldsymbol{\theta}}(R \circ \mathcal{L} \circ h)(\boldsymbol{\theta}), \boldsymbol{\theta}' - \boldsymbol{\theta}\rangle \\
& + \|\nabla_h(R \circ \mathcal{L})(h(\boldsymbol{\theta}))\|_{2,1}\, O\left(\frac{L^2\omega^{1/3}\sqrt{m\log(m)}}{\sqrt{d}}\right) \|\boldsymbol{\theta}' - \boldsymbol{\theta}\|_{2,\infty} \\
& + O\left(\left(\frac{\beta C(\mathcal{L})^2}{n\rho} + 2C(\nabla \mathcal{L})\right)\frac{nL^2 m}{d}\right) \|\boldsymbol{\theta}' - \boldsymbol{\theta}\|_{2,\infty}^2
\end{aligned}
\tag{56}
$$

where for all layer $l \in \{1, \ldots, L\}$, $\boldsymbol{\theta}_l$ is the vector of parameters for layer $l$, and

$$
\|\boldsymbol{\theta}' - \boldsymbol{\theta}\|_{2,\infty} = \max_l \|\boldsymbol{\theta}'_l - \boldsymbol{\theta}_l\|_2
$$

$$
\|\boldsymbol{\theta}' - \boldsymbol{\theta}\|_{2,\infty}^2 = \left(\max_l \|\boldsymbol{\theta}'_l - \boldsymbol{\theta}_l\|_2^2\right)^2 = \max_l \|\boldsymbol{\theta}'_l - \boldsymbol{\theta}_l\|_2^2
$$

$$
\begin{aligned}
\|\nabla_h(R \circ \mathcal{L})(h(\boldsymbol{\theta}))\|_{2,1} &= \sum_{i=1}^{n} \|\nabla_{h_i}(R \circ \mathcal{L})(h(\boldsymbol{\theta}))\|_2 \\
&= \sum_{i=1}^{n} \left\|\bar{p}_i(\mathcal{L}(h(\boldsymbol{\theta}))) \nabla_{h_i} \underset{i}{\mathcal{L}}(h_i(\boldsymbol{\theta}))\right\|_2 \quad \text{(chain rule (54))}
\end{aligned}
$$

To compare this semi-smoothness result to the one in (Allen-Zhu et al., 2019, Theorem 4), let us first remark that

$$
\|\nabla_h(R \circ \mathcal{L})(h(\boldsymbol{\theta}))\|_{2,1} \leq \sqrt{n} \|\nabla_h(R \circ \mathcal{L})(h(\boldsymbol{\theta}))\|_{2,2}
$$

As a result, our result is analogous to (Allen-Zhu et al., 2019, Theorem 4), up to an additional multiplicative factor $\left(\frac{\beta C(\mathcal{L})^2}{n\rho} + 2C(\nabla \mathcal{L})\right)$ in the last term of the right-hand side. It is worth noting that there is also implicitly an additional multiplicative factor $C(\nabla \mathcal{L})$ in Theorem 3 of (Allen-Zhu et al., 2019) since (Allen-Zhu et al., 2019) make the assumption that $C(\nabla \mathcal{L}) = 1$ (see Allen-Zhu et al., 2019, Appendix A).

Let $\boldsymbol{\theta}, \boldsymbol{\theta}' \in (\mathbb{R}^{m \times m})^L$ verifying the conditions of Theorem D.1.

Let $A = R(\mathcal{L}(h(\boldsymbol{\theta}'))) - R(\mathcal{L}(h(\boldsymbol{\theta})) - \langle \nabla_{\boldsymbol{\theta}}(R \circ \mathcal{L} \circ h)(\boldsymbol{\theta}), \boldsymbol{\theta}' - \boldsymbol{\theta}\rangle$ , the quantity we want to bound.

Using (53) for $z = h(\boldsymbol{\theta})$ and $z' = h(\boldsymbol{\theta}')$, we obtain

$$
\begin{aligned}
A \leq \frac{1}{2} \left( \frac{\beta C(\mathcal{L})^2}{n\rho} + 2C(\nabla \mathcal{L}) \right) & \|h(\boldsymbol{\theta}') - h(\boldsymbol{\theta})\|_2^2 \\
& + \langle \nabla_h (R \circ \mathcal{L})(h(\boldsymbol{\theta})), h(\boldsymbol{\theta}') - h(\boldsymbol{\theta}) \rangle \\
& - \langle \nabla_{\boldsymbol{\theta}} (R \circ \mathcal{L} \circ h)(\boldsymbol{\theta}), \boldsymbol{\theta}' - \boldsymbol{\theta} \rangle
\end{aligned}
\tag{57}
$$

Then using the chain rule (55)

$$
\begin{aligned}
A \leq \frac{1}{2} \left( \frac{\beta C(\mathcal{L})^2}{n\rho} + 2C(\nabla \mathcal{L}) \right) & \|h(\boldsymbol{\theta}') - h(\boldsymbol{\theta})\|_2^2 \\
& + \sum_{i=1}^{n} \langle \nabla_{h_i} (R \circ \mathcal{L})(h(\boldsymbol{\theta})), h_i(\boldsymbol{\theta}') - h_i(\boldsymbol{\theta}) - (\nabla_{\boldsymbol{\theta}} h_i(\boldsymbol{\theta}))^T (\boldsymbol{\theta}' - \boldsymbol{\theta}) \rangle
\end{aligned}
\tag{58}
$$

For all $i \in \{1, \ldots, n\}$, let us denote $\overset{\smile}{loss}_i := \nabla_{h_i}(R \circ \mathcal{L})(h(\boldsymbol{\theta}))$ to match the notations used in (Allen-Zhu et al., 2019) for the derivative of the loss with respect to the output of the network for example i of the training set.

With this notation, we obtain exactly equation (11.3) in (Allen-Zhu et al., 2019) up to the multiplicative factor $\left( \frac{\beta C(\mathcal{L})^2}{n\rho} + 2C(\nabla \mathcal{L}) \right)$ for the distributionally robust loss.

From there the proof of Theorem 4 in (Allen-Zhu et al., 2019) being independent to the formula for $\overset{\smile}{loss}_i$, we can conclude the proof of our Theorem D.1 (as in Allen-Zhu et al., 2019, Appendix A).

### D.7.2 GRADIENT BOUNDS FOR THE DISTRIBUTIONALLY ROBUST LOSS

We prove the following lemma which is a generalization of Theorem 3 in (Allen-Zhu et al., 2019) for the distributionally robust loss (9).

**Theorem D.2** (Gradient Bounds for the Distributionally Robust Loss)**.**

*Let* $\omega \in O\left( \frac{\delta^{3/2}}{n^{9/2}L^6 \log^3(m)} \right)$, *and* $\boldsymbol{\theta}^{(0)}$ *being initialized randomly as described in assumption B.1. With probability as least* $1 - \exp\left(-\Omega(m\omega^{3/2}L)\right)$ *over the initialization, we have for all* $\boldsymbol{\theta} \in (\mathbb{R}^{m \times m})^L$ *with* $\left\| \boldsymbol{\theta} - \boldsymbol{\theta}^{(0)} \right\|_2 \leq \omega$

$$
\begin{aligned}
& \forall i \in \{1, \ldots, n\}, \ \forall l \in \{1, \ldots, L\}, \ \forall \hat{\mathcal{L}} \in \mathbb{R}^n \\
& \left\| \bar{p}_i(\hat{\mathcal{L}}) \nabla_{\boldsymbol{\theta}_l} (\underset{i}{\mathcal{L}} \circ h_i)(\boldsymbol{\theta}) \right\|_2^2 \leq O\left( \frac{m}{d} \left\| \bar{p}_i(\hat{\mathcal{L}}) \nabla_{h_i} \underset{i}{\mathcal{L}}(h_i(\boldsymbol{\theta})) \right\|_2^2 \right) \\
& \forall l \in \{1, \ldots, L\}, \ \forall \hat{\mathcal{L}} \in \mathbb{R}^n \\
& \left\| \sum_{i=1}^{n} \bar{p}_i(\hat{\mathcal{L}}) \nabla_{\boldsymbol{\theta}_l} (\underset{i}{\mathcal{L}} \circ h_i)(\boldsymbol{\theta}) \right\|_2^2 \leq O\left( \frac{mn}{d} \sum_{i=1}^{n} \left\| \bar{p}_i(\hat{\mathcal{L}}) \nabla_{h_i} \underset{i}{\mathcal{L}}(h_i(\boldsymbol{\theta})) \right\|_2^2 \right) \\
& \left\| \sum_{i=1}^{n} \bar{p}_i(\hat{\mathcal{L}}) \nabla_{\boldsymbol{\theta}_L} (\underset{i}{\mathcal{L}} \circ h_i)(\boldsymbol{\theta}) \right\|_2^2 \geq \Omega\left( \frac{m\delta}{dn^2} \sum_{i=1}^{n} \left\| \bar{p}_i(\hat{\mathcal{L}}) \nabla_{h_i} \underset{i}{\mathcal{L}}(h_i(\boldsymbol{\theta})) \right\|_2^2 \right)
\end{aligned}
\tag{59}
$$

It is worth noting that the loss vector $\hat{\mathcal{L}}$ used for computing the robust probabilities $\bar{p}(\hat{\mathcal{L}}) = \left( \bar{p}_i(\hat{\mathcal{L}}) \right)_{i=1}^{n}$ does not have to be equal to $\mathcal{L}(h(\boldsymbol{\theta}))$.

We will use this for the proof of the Robust SGD with stale loss history.

The adaptation of the proof of Theorem 3 in (Allen-Zhu et al., 2019) is straightforward.

Let $\boldsymbol{\theta} \in (\mathbb{R}^{m \times m})^L$ satisfying the conditions of Theorem D.2, and $\hat{\mathcal{L}} \in \mathbb{R}^n$.

Let us denote $v := \left( \bar{p}_i(\hat{\mathcal{L}}) \nabla_{h_i} \mathcal{L}_i(h_i(\boldsymbol{\theta})) \right)_{i=1}^n$, applying the proof of Theorem 3 in (Allen-Zhu et al., 2019) to our $v$ gives:

$$\forall i \in \{1, \ldots, n\}, \ \forall l \in \{1, \ldots, L\},$$

$$\left\| \bar{p}_i(\hat{\mathcal{L}}) \nabla_{\boldsymbol{\theta}_l} (\mathcal{L} \circ h_i)(\boldsymbol{\theta}) \right\|_2^2 \leq O \left( \frac{m}{d} \left\| \bar{p}_i(\hat{\mathcal{L}}) \nabla_{h_i} \mathcal{L}_i(h_i(\boldsymbol{\theta})) \right\|_2^2 \right)$$

$$\forall l \in \{1, \ldots, L\}, \ \forall \hat{\mathcal{L}} \in \mathbb{R}^n$$

$$\left\| \sum_{i=1}^n \bar{p}_i(\hat{\mathcal{L}}) \nabla_{\boldsymbol{\theta}_l} (\mathcal{L} \circ h_i)(\boldsymbol{\theta}) \right\|_2^2 \leq O \left( \frac{mn}{d} \sum_{i=1}^n \left\| \bar{p}_i(\hat{\mathcal{L}}) \nabla_{h_i} \mathcal{L}_i(h_i(\boldsymbol{\theta})) \right\|_2^2 \right)$$

$$\left\| \sum_{i=1}^n \bar{p}_i(\hat{\mathcal{L}}) \nabla_{\boldsymbol{\theta}_L} (\mathcal{L} \circ h_i)(\boldsymbol{\theta}) \right\|_2^2 \geq \Omega \left( \frac{m\delta}{dn} \max_i \left( \left\| \bar{p}_i(\hat{\mathcal{L}}) \nabla_{h_i} \mathcal{L}_i(h_i(\boldsymbol{\theta})) \right\|_2^2 \right) \right)$$

In addition

$$\max_i \left( \left\| \bar{p}_i(\hat{\mathcal{L}}) \nabla_{h_i} \mathcal{L}_i(h_i(\boldsymbol{\theta})) \right\|_2^2 \right) \geq \frac{1}{n} \sum_{i=1}^n \left\| \bar{p}_i(\hat{\mathcal{L}}) \nabla_{h_i} \mathcal{L}_i(h_i(\boldsymbol{\theta})) \right\|_2^2$$

This allows us to conclude the proof of our Theorem D.2.

### D.7.3 Convergence of Robust SGD with Exact Loss History

We can now prove Theorem B.1.

**Theorem D.3** (Convergence of Robust SGD with exact Loss History – Restated from Theorem B.1). *Suppose batch size $1 \leq b \leq n$, number of hidden units $m \geq \Omega(d\epsilon^{-1} \times \text{poly}(n, L, \delta^{-1}))$, and $\delta \geq O\left(\frac{1}{L}\right)$. Let $\epsilon > 0$, and the learning rate be $\eta_{exact} = \Theta \left( \frac{\alpha n^2 \rho}{\beta C(\mathcal{L})^2 + 2n\rho C(\nabla \mathcal{L})} \times \frac{b\delta d}{\text{poly}(n, L) m \log^2(m)} \right)$, with probability at least $1 - \exp\left(-\Omega(\log^2(m))\right)$ over the randomness of the initialization and the mini-batches, Robust SGD with exact loss vector finds $\|\nabla_{\boldsymbol{\theta}}(R \circ f \circ h)(\boldsymbol{\theta})\| \leq \epsilon$ after $T = O\left(\frac{Ln^3}{\eta\delta\epsilon^2}\right)$ iterations.*

Similarly to the proof of the convergence of SGD for the mean loss (4) (Theorem 2 in (Allen-Zhu et al., 2019)), the convergence of SGD for the distributionally robust loss (9) will mainly rely on the semi-smoothness property (Theorem D.1) and the gradient bound (Theorem D.2) that we have proved previously for the distributionally robust loss.

Let $\boldsymbol{\theta} \in (\mathbb{R}^{m \times m})^L$ satisfying the conditions of Theorem B.1, and $\hat{\mathcal{L}}$ be the exact loss history at $\boldsymbol{\theta}$, i.e.

$$\hat{\mathcal{L}} = \left( \mathcal{L}(h_i(\boldsymbol{\theta})) \right)_{i=1}^n \tag{60}$$

For the batch size $b \in \{1, \ldots, n\}$, let $S = \{i_j\}_{j=1}^b$ a batch of indices drawn from $\bar{p}(\hat{\mathcal{L}})$ without replacement, i.e.

$$\forall j \in \{1, \ldots b\}, \ i_j \overset{\text{i.i.d.}}{\sim} \bar{p}(\hat{\mathcal{L}}) \tag{61}$$

Let $\boldsymbol{\theta}' \in (\mathbb{R}^{m \times m})^L$ be the values of the parameters after a stochastic gradient descent step at $\boldsymbol{\theta}$ for the batch $S$, i.e.

$$\boldsymbol{\theta}' = \boldsymbol{\theta} - \eta \frac{1}{b} \sum_{i \in S} \nabla_{\boldsymbol{\theta}} (\mathcal{L} \circ h_i)(\boldsymbol{\theta}) \tag{62}$$

where $\eta > 0$ is the learning rate.

Assuming that $\boldsymbol{\theta}$ and $\boldsymbol{\theta}'$ satisfies the conditions of Theorem D.1, we obtain

$$
\begin{aligned}
R(\mathcal{L}(h(\boldsymbol{\theta}'))) \leq & R(\mathcal{L}(h(\boldsymbol{\theta}))) - \eta \langle \nabla_{\boldsymbol{\theta}}(R \circ \mathcal{L} \circ h)(\boldsymbol{\theta}), \frac{1}{b} \sum_{i \in S} \nabla_{\boldsymbol{\theta}}(\underset{i}{\mathcal{L}} \circ h_i)(\boldsymbol{\theta}) \rangle \\
& + \eta \sqrt{n} \left\| \nabla_h(R \circ \mathcal{L})(h(\boldsymbol{\theta})) \right\|_{2,2} O\left( \frac{L^2 \omega^{1/3} \sqrt{m \log(m)}}{\sqrt{d}} \right) \left\| \frac{1}{b} \sum_{i \in S} \nabla_{\boldsymbol{\theta}}(\underset{i}{\mathcal{L}} \circ h_i)(\boldsymbol{\theta}) \right\|_{2,\infty} \\
& + \eta^2 O\left( \left( \frac{\beta C(\mathcal{L})^2}{n\rho} + 2C(\nabla \mathcal{L}) \right) \frac{nL^2 m}{d} \right) \left\| \frac{1}{b} \sum_{i \in S} \nabla_{\boldsymbol{\theta}}(\underset{i}{\mathcal{L}} \circ h_i)(\boldsymbol{\theta}) \right\|_{2,\infty}^2
\end{aligned}
\tag{63}
$$

where we refer to (55) for the form of $\nabla_{\boldsymbol{\theta}}(R \circ \mathcal{L} \circ h)(\boldsymbol{\theta})$ and to (54) for the form of $\nabla_h(R \circ \mathcal{L})(h(\boldsymbol{\theta}))$.

In addition, we make the assumption that for the set of values of $\boldsymbol{\theta}$ considered the hardness weighted sampling probabilities admit an upper-bound

$$
\alpha = \min_{\boldsymbol{\theta}} \min_i \bar{p}_i(\mathcal{L}(\boldsymbol{\theta})) > 0
\tag{64}
$$

Which is always satisfied under assumption B.2 for Kullback-Leibler $\phi$-divergence, and for any $\phi$-divergence satisfying assumption 3.1 with a robustness parameter $\beta$ small enough.

Let $\mathbb{E}_S$ be the expectation with respect to $S$. Applying $\mathbb{E}_S$ to (63), we obtain

$$
\begin{aligned}
& \mathbb{E}_S \left[ R(\mathcal{L}(h(\boldsymbol{\theta}'))) \right] \\
& \leq R(\mathcal{L}(h(\boldsymbol{\theta}))) - \eta \left\| \nabla_{\boldsymbol{\theta}}(R \circ \mathcal{L} \circ h)(\boldsymbol{\theta}) \right\|_{2,2}^2 \\
& + \eta \left\| \nabla_h(R \circ \mathcal{L})(h(\boldsymbol{\theta})) \right\|_{2,2} O\left( \frac{nL^2 \omega^{1/3} \sqrt{m \log(m)}}{\sqrt{d}} \right) \sqrt{\sum_{i=1}^n \max_l \left\| \bar{p}_i(\hat{\mathcal{L}}) \nabla_{\boldsymbol{\theta}_l}(\underset{i}{\mathcal{L}} \circ h_i)(\boldsymbol{\theta}) \right\|^2} \\
& + \eta^2 O\left( \left( \frac{\beta C(\mathcal{L})^2}{n\rho} + 2C(\nabla \mathcal{L}) \right) \frac{nL^2 m}{d} \right) \frac{1}{\alpha} \sum_{i=1}^n \max_l \left\| \bar{p}_i(\hat{\mathcal{L}}) \nabla_{\boldsymbol{\theta}_l}(\underset{i}{\mathcal{L}} \circ h_i)(\boldsymbol{\theta}) \right\|^2
\end{aligned}
\tag{65}
$$

where we have used the following results:

- For any integer $k \geq 1$, and all $(a_i)_{i=1}^n \in \left( \mathbb{R}^k \right)^n$, we have (see the proof in D.7.4)

$$
\mathbb{E}_S \left[ \frac{1}{b} \sum_{i \in S} a_i \right] = \mathbb{E}_{\bar{p}(\hat{\mathcal{L}})} [a_i]
\tag{66}
$$

- Using (66) for $(a_i)_{i=1}^n = (\nabla_{\boldsymbol{\theta}}(\mathcal{L}_i \circ h_i)(\boldsymbol{\theta}))_{i=1}^n$, and the chain rule (55)

$$
\mathbb{E}_S \left[ \frac{1}{b} \sum_{i \in S} \nabla_{\boldsymbol{\theta}}(\underset{i}{\mathcal{L}} \circ h_i)(\boldsymbol{\theta}) \right] = \sum_{i=1}^n \bar{p}_i(\hat{\mathcal{L}}) \nabla_{\boldsymbol{\theta}}(\underset{i}{\mathcal{L}} \circ h_i)(\boldsymbol{\theta}) = \nabla_{\boldsymbol{\theta}}(R \circ \mathcal{L} \circ h)(\boldsymbol{\theta})
\tag{67}
$$

- Using the triangular inequality

$$
\left\| \frac{1}{b} \sum_{i \in S} \nabla_{\boldsymbol{\theta}}(\underset{i}{\mathcal{L}} \circ h_i)(\boldsymbol{\theta}) \right\|_{2,\infty} \leq \frac{1}{b} \sum_{i \in S} \left\| \nabla_{\boldsymbol{\theta}}(\underset{i}{\mathcal{L}} \circ h_i)(\boldsymbol{\theta}) \right\|_{2,\infty}
\tag{68}
$$

And using (66) for $(a_i)_{i=1}^n = \left( \left\| \nabla_{\boldsymbol{\theta}}(\mathcal{L}_i \circ h_i)(\boldsymbol{\theta}) \right\|_{2,\infty} \right)_{i=1}^n$,

$$
\begin{aligned}
\mathbb{E}_S \left[ \left\| \frac{1}{b} \sum_{i \in S} \nabla_{\boldsymbol{\theta}}(\underset{i}{\mathcal{L}} \circ h_i)(\boldsymbol{\theta}) \right\|_{2,\infty} \right] & \leq \sum_{i=1}^n \bar{p}_i(\hat{\mathcal{L}}) \left\| \nabla_{\boldsymbol{\theta}}(\underset{i}{\mathcal{L}} \circ h_i)(\boldsymbol{\theta}) \right\|_{2,\infty} \\
& \leq \sum_{i=1}^n \max_l \left\| \nabla_{\boldsymbol{\theta}_l}(\bar{p}_i(\hat{\mathcal{L}}) \underset{i}{\mathcal{L}} \circ h_i)(\boldsymbol{\theta}) \right\|_2 \\
& \leq \sqrt{n} \sqrt{\sum_{i=1}^n \max_l \left\| \nabla_{\boldsymbol{\theta}_l}(\bar{p}_i(\hat{\mathcal{L}}) \underset{i}{\mathcal{L}} \circ h_i)(\boldsymbol{\theta}) \right\|_2^2}
\end{aligned}
\tag{69}
$$

where we have used Cauchy-Schwarz inequality for the last inequality.

- Using (68) and the convexity of the function $x \mapsto x^2$

$$\left\| \frac{1}{b} \sum_{i \in S} \nabla_{\boldsymbol{\theta}} (\mathcal{L} \circ h_i)(\boldsymbol{\theta}) \right\|_{2,\infty}^2 \leq \frac{1}{b} \sum_{i \in S} \left\| \nabla_{\boldsymbol{\theta}} (\mathcal{L} \circ h_i)(\boldsymbol{\theta}) \right\|_{2,\infty}^2 \tag{70}$$

And using (66) for $(a_i)_{i=1}^n = \left( \| \nabla_{\boldsymbol{\theta}} (\mathcal{L}_i \circ h_i)(\boldsymbol{\theta}) \|_{2,\infty}^2 \right)_{i=1}^n$,

$$\mathbb{E}_S \left[ \left\| \frac{1}{b} \sum_{i \in S} \nabla_{\boldsymbol{\theta}} (\mathcal{L} \circ h_i)(\boldsymbol{\theta}) \right\|_{2,\infty}^2 \right] \leq \sum_{i=1}^n \bar{p}_i(\hat{\mathcal{L}}) \left\| \nabla_{\boldsymbol{\theta}} (\mathcal{L} \circ h_i)(\boldsymbol{\theta}) \right\|_{2,\infty}^2$$

$$\leq \sum_{i=1}^n \frac{1}{\bar{p}_i(\hat{\mathcal{L}})} \max_l \left\| \nabla_{\boldsymbol{\theta}_l} (\bar{p}_i(\hat{\mathcal{L}}) \, \mathcal{L} \circ h_i)(\boldsymbol{\theta}) \right\|_2^2 \tag{71}$$

$$\leq \frac{1}{\alpha} \sum_{i=1}^n \max_l \left\| \nabla_{\boldsymbol{\theta}_l} (\bar{p}_i(\hat{\mathcal{L}}) \, \mathcal{L} \circ h_i)(\boldsymbol{\theta}) \right\|_2^2$$

**Important Remark:** It is worth noting the apparition of $\alpha$ (64) in (71). If we were using a uniform sampling as for ERM (i.e. for DRO in the limit $\beta \to 0$), we would have $\alpha = \frac{1}{n}$. So although our inequality (71) may seem brutal, it is consistent with equation (13.2) in (Allen-Zhu et al., 2019) and the corresponding inequality in the case of ERM.

The rest of the proof of convergence will consist in proving that $\eta \| \nabla_{\boldsymbol{\theta}} (R \circ \mathcal{L} \circ h)(\boldsymbol{\theta}) \|_{2,2}^2$ dominates the two last terms in (63). As a result, we can already state that either the robustness parameter $\beta$, or the learning rate $\eta$ will have to be small enough to control $\alpha$. This is consistent with what we observed in our experiments.

Indeed, combining (63) with the chain rule (55), and the gradient bound Theorem D.2 where we use our $\hat{\mathcal{L}}$ defined in (60)

$$\mathbb{E}_S \left[ R(\mathcal{L}(h(\boldsymbol{\theta}'))) \right] \leq R(\mathcal{L}(h(\boldsymbol{\theta})) - \Omega \left( \frac{\eta m \delta}{dn^2} \right) \sum_{i=1}^n \left\| \bar{p}_i(\hat{\mathcal{L}}) \nabla_{h_i} \mathcal{L}(h_i(\boldsymbol{\theta})) \right\|_2^2$$

$$+ \eta O \left( \frac{nL^2 \omega^{1/3} \sqrt{m \log(m)}}{\sqrt{d}} \right) O \left( \sqrt{\frac{m}{d}} \right) \sum_{i=1}^n \left\| \bar{p}_i(\hat{\mathcal{L}}) \nabla_{h_i} \mathcal{L}(h_i(\boldsymbol{\theta})) \right\|_2^2$$

$$+ \eta^2 O \left( \left( \frac{\beta C(\mathcal{L})^2}{n \rho} + 2C(\nabla \mathcal{L}) \right) \frac{nL^2 m}{d} \right) O \left( \frac{m}{d\alpha} \right) \sum_{i=1}^n \left\| \bar{p}_i(\hat{\mathcal{L}}) \nabla_{h_i} \mathcal{L}(h_i(\boldsymbol{\theta})) \right\|_2^2$$

$$\leq R(\mathcal{L}(h(\boldsymbol{\theta})) - \Omega \left( \frac{\eta m \delta}{dn^2} \right) \sum_{i=1}^n \left\| \bar{p}_i(\hat{\mathcal{L}}) \nabla_{h_i} \mathcal{L}(h_i(\boldsymbol{\theta})) \right\|_2^2$$

$$+ O \left( \frac{\eta nL^2 m \omega^{1/3} \sqrt{\log(m)}}{d} + K \frac{\eta^2 (n/\alpha) L^2 m^2}{d^2} \right) \sum_{i=1}^n \left\| \bar{p}_i(\hat{\mathcal{L}}) \nabla_{h_i} \mathcal{L}(h_i(\boldsymbol{\theta})) \right\|_2^2 \tag{72}$$

where we have used

$$K := \frac{\beta C(\mathcal{L})^2}{n \rho} + 2C(\nabla \mathcal{L}) \tag{73}$$

There are only two differences with equation (13.2) in (Allen-Zhu et al., 2019):

- in the last fraction we have $n/\alpha$ instead of $n^2$ (see remark D.7.3 for more details), and an additional multiplicative term $K$. So in total, this term differs by a multiplicative factor $\frac{\alpha n}{K}$ from the analogous term in the proof of (Allen-Zhu et al., 2019).

- we have $\sum_{i=1}^n \left\| \bar{p}_i(\hat{\mathcal{L}}) \nabla_{h_i} \mathcal{L}_i(h_i(\boldsymbol{\theta})) \right\|_2^2$ instead of $F(\mathbf{W}^{(t)})$. In fact they are analogous since in equation (13.2) in (Allen-Zhu et al., 2019), $F(\mathbf{W}^{(t)})$ is the squared norm of the

mean loss for the $L^2$ loss. We don't make such a strong assumption on the choice of $\mathcal{L}$ (see assumption B.2). It is worth noting that the same analogy is used in (Allen-Zhu et al., 2019, Appendix A) where they extend their result to the mean loss with other objective function than the $L^2$ loss.

Our choice of learning rate in Theorem B.2 can be rewritten as

$$
\begin{aligned}
\eta_{exact} &= \Theta\left(\frac{\alpha n^2 \rho}{\beta C(\mathcal{L})^2 + 2n\rho C(\nabla \mathcal{L})} \times \frac{b\delta d}{\text{poly}(n,L)m\log^2(m)}\right) \\
&= \Theta\left(\frac{\alpha n}{K} \times \frac{b\delta d}{\text{poly}(n,L)m\log^2(m)}\right) \\
&\leq \frac{\alpha n}{K} \times \eta'
\end{aligned}
\tag{74}
$$

And we also have

$$
\eta_{exact} \leq \eta' \tag{75}
$$

where $\eta'$ is the learning rate chosen in the proof of Theorem 2 in (Allen-Zhu et al., 2019). We refer the reader to (Allen-Zhu et al., 2019) for the details of the constant in "$\Theta$" and the exact form of the polynom $\text{poly}(n,L)$.

As a result, for $\eta = \eta_{exact}$, the term $\Omega\left(\frac{\eta m\delta}{dn^2}\right)$ dominates the other term of the right-hand side of inequality (72) as in the proof of Theorem 2 in (Allen-Zhu et al., 2019).

This implies that the conditions of Theorem D.2 are satisfied for all $\boldsymbol{\theta}^{(t)}$, and that we have for all iteration $t > 0$

$$
\mathbb{E}_{S_t}\left[R(\mathcal{L}(h(\boldsymbol{\theta}^{(t+1)})))\right] \leq R(\mathcal{L}(h(\boldsymbol{\theta}^{(t)}))) - \Omega\left(\frac{\eta m\delta}{dn^2}\right)\sum_{i=1}^{n}\left\|\bar{p}_i(\hat{\mathcal{L}})\nabla_{h_i}\underset{i}{\mathcal{L}}(h_i(\boldsymbol{\theta}^{(t)}))\right\|_2^2 \tag{76}
$$

And using a result in Appendix A of (Allen-Zhu et al., 2019), since under assumption B.2 the distributionally robust loss is non-convex and bounded, we obtain for all $\epsilon' > 0$

$$
\left\|\nabla_h(R \circ \mathcal{L})(h(\boldsymbol{\theta}^{(T)}))\right\|_{2,2} \leq \epsilon' \quad \text{if} \quad T = O\left(\frac{dn^2}{\eta\delta m\epsilon'^2}\right) \tag{77}
$$

where according to (54)

$$
\left\|\nabla_h(R \circ \mathcal{L})(h(\boldsymbol{\theta}^{(T)}))\right\|_{2,2} = \sum_{i=1}^{n}\left\|\bar{p}_i(\hat{\mathcal{L}})\nabla_{h_i}\underset{i}{\mathcal{L}}(h_i(\boldsymbol{\theta}^{(t)}))\right\|_2^2 \tag{78}
$$

However, we are interested in a bound on $\left\|\nabla_{\boldsymbol{\theta}}(R \circ \mathcal{L} \circ h)(\boldsymbol{\theta}^{(T)})\right\|_{2,2}$, rather than $\left\|\nabla_h(R \circ \mathcal{L})(h(\boldsymbol{\theta}^{(T)}))\right\|_{2,2}$.

Using the gradient bound of Theorem D.2 and the chain rules (55) and (54)

$$
\left\|\nabla_{\boldsymbol{\theta}}(R \circ \mathcal{L} \circ h)(\boldsymbol{\theta}^{(T)})\right\|_{2,2} \leq c_1\sqrt{\frac{Lmn}{d}}\left\|\nabla_h(R \circ \mathcal{L})(h(\boldsymbol{\theta}^{(T)}))\right\|_{2,2} \tag{79}
$$

where $c_1 > 0$ is the constant hidden in $O\left(\sqrt{\frac{Lmn}{d}}\right)$.

So with $\epsilon' = \frac{1}{c_1}\sqrt{\frac{d}{Lmn}}\epsilon$, we finally obtain

$$
\begin{aligned}
\left\|\nabla_{\boldsymbol{\theta}}(R \circ \mathcal{L} \circ h)(\boldsymbol{\theta}^{(T)})\right\|_{2,2} &\leq c_1\sqrt{\frac{Lmn}{d}}\left\|\nabla_h(R \circ \mathcal{L})(h(\boldsymbol{\theta}^{(T)}))\right\|_{2,2} \\
&\leq c_1\sqrt{\frac{Lmn}{d}}\epsilon' \\
&\leq \epsilon
\end{aligned}
\tag{80}
$$

If

$$
T = O\left(\frac{dn^2}{\eta\delta m\epsilon'^2}\right) = O\left(\frac{dn^2}{\eta\delta m}\frac{Lmn}{d\epsilon^2}\right) = O\left(\frac{Ln^3}{\eta\delta\epsilon^2}\right) \tag{81}
$$

which concludes the proof.

### D.7.4 Proof of technical lemma 1

For any integer $k \geq 1$, and all $(a_i)_{i=1}^n \in (\mathbb{R}^k)^n$, we have

$$
\begin{aligned}
\mathbb{E}_S\left[\frac{1}{b}\sum_{i \in S} a_i\right] &= \sum_{1 \leq i_1, \ldots, i_b \leq n}\left[\left(\prod_{k=1}^n \bar{p}_{i_k}(\hat{\mathcal{L}})\right)\frac{1}{b}\sum_{j=1}^b a_{i_j}\right] \\
&= \frac{1}{b}\sum_{1 \leq i_1, \ldots, i_b \leq n}\left[\sum_{j=1}^b \bar{p}_{i_j}(\hat{\mathcal{L}})\, a_{i_j}\left(\prod_{\substack{k=1 \\ k \neq j}}^n \bar{p}_{i_k}(\hat{\mathcal{L}})\right)\right] \\
&= \frac{1}{b}\sum_{j=1}^b\left[\sum_{1 \leq i_1, \ldots, i_b \leq n} \bar{p}_{i_j}(\hat{\mathcal{L}})\, a_{i_j}\left(\prod_{\substack{k=1 \\ k \neq j}}^n \bar{p}_{i_k}(\hat{\mathcal{L}})\right)\right] \\
&= \frac{1}{b}\sum_{j=1}^b\left[\left(\sum_{i_j=1}^n \bar{p}_{i_j}(\hat{\mathcal{L}})\, a_{i_j}\right)\prod_{\substack{k=1 \\ k \neq j}}^n\left(\sum_{i_k=1}^n \bar{p}_{i_k}(\hat{\mathcal{L}})\right)\right] \\
&= \frac{1}{b}\sum_{j=1}^b\left(\sum_{i=1}^n \bar{p}_i(\hat{\mathcal{L}})\, a_i\right) \\
&= \sum_{i=1}^n \bar{p}_i(\hat{\mathcal{L}})\, a_i \\
&= \mathbb{E}_{\bar{p}(\hat{\mathcal{L}})}\left[a_i\right]
\end{aligned}
\tag{82}
$$

## D.8 Convergence of Robust SGD with Stale Loss History

The proof of the convergence of Algorithm 4.1 under the conditions of Theorem B.2 follows the same structure as the proof of the convergence of Robust SGD with exact loss history D.7.3. We will reuse the intermediate results of D.7.3 when possible and focus on the differences between the two proofs due to the inexactness of the loss history.

Let an iteration number $t$, so that the warm-up of Algorithm 4.1 is already over at $t$.

Let $\boldsymbol{\theta}^{(t)} \in (\mathbb{R}^{m \times m})^L$ the parameters of the deep neural network at iteration $t$.

We define the stale loss history at iteration $t$ as

$$
\hat{\mathcal{L}} = \left(\underset{i}{\mathcal{L}}(h_i(\boldsymbol{\theta}^{(t_i(t))}))\right)_{i=1}^n
\tag{83}
$$

where for all $i$, $t_i(t) < t$ corresponds to the latest iteration before $t$ at which the loss for example $i$ has been updated. Or equivalently, it corresponds to the last iteration before $t$ when example $i$ was drawn to be part of a mini-batch.

Thanks to the warm-up stage of Algorithm 4.1, it is guaranteed that the loss value of every example has been computed at least once before we start using the adaptive sampling. As a result, for all iteration after the warm-up, the stale loss history $\hat{\mathcal{L}}$ is well defined.

We also define the exact loss history that is unknown in Algorithm 4.1, as

$$
\check{\mathcal{L}} = \left(\underset{i}{\mathcal{L}}(h_i(\boldsymbol{\theta}^{(t)}))\right)_{i=1}^n
\tag{84}
$$

**Remark on the warm-up stage of Algorithm 4.1:** The iterations performed during the warm-up stage amounts to classic SGD to minimize the mean loss (4). As a result, the convergence results of (Allen-Zhu et al., 2019, Theorem 2) apply during the warm-up. This guarantees that the condition on $\boldsymbol{\theta}$ of Theorem B.2 remains satisfied during the warm-up if it was satisfied by the initial parameters.

Similarly to (62) we define

$$\boldsymbol{\theta}^{(t+1)} = \boldsymbol{\theta}^{(t)} - \eta \frac{1}{b} \sum_{i \in S} \nabla_{\boldsymbol{\theta}} (\underset{i}{\mathcal{L}} \circ h_i)(\boldsymbol{\theta}^{(t)}) \tag{85}$$

and using Theorem D.1, similarly to (63), we obtain

$$
\begin{aligned}
R(\mathcal{L}(h(\boldsymbol{\theta}^{(t+1)}))) \leq & R(\mathcal{L}(h(\boldsymbol{\theta}^{(t)}))) - \eta \langle \nabla_{\boldsymbol{\theta}}(R \circ \mathcal{L} \circ h)(\boldsymbol{\theta}^{(t)}), \frac{1}{b} \sum_{i \in S} \nabla_{\boldsymbol{\theta}}(\underset{i}{\mathcal{L}} \circ h_i)(\boldsymbol{\theta}^{(t)}) \rangle \\
& + \eta \left\| \nabla_h (R \circ \mathcal{L})(h(\boldsymbol{\theta}^{(t)})) \right\|_{1,2} O\left( \frac{L^2 \omega^{1/3} \sqrt{m \log(m)}}{\sqrt{d}} \right) \left\| \frac{1}{b} \sum_{i \in S} \nabla_{\boldsymbol{\theta}}(\underset{i}{\mathcal{L}} \circ h_i)(\boldsymbol{\theta}^{(t)}) \right\|_{2,\infty} \\
& + \eta^2 O\left( \left( \frac{\beta C(\mathcal{L})^2}{n\rho} + 2C(\nabla \mathcal{L}) \right) \frac{nL^2 m}{d} \right) \left\| \frac{1}{b} \sum_{i \in S} \nabla_{\boldsymbol{\theta}}(\underset{i}{\mathcal{L}} \circ h_i)(\boldsymbol{\theta}^{(t)}) \right\|_{2,\infty}^2
\end{aligned}
\tag{86}
$$

We can still define $\alpha$ as in (64)

$$\alpha = \min_{\boldsymbol{\theta}} \min_i \bar{p}_i(\mathcal{L}(\boldsymbol{\theta})) > 0 \tag{87}$$

where we are guaranteed that $\alpha > 0$ under assumptions B.1.

Since Theorem D.2 is independent to the choice of $\hat{\mathcal{L}}$, taking the expectation with respect to $S$, similarly to (72), we obtain

$$
\begin{aligned}
\mathbb{E}_S \left[ R(\mathcal{L}(h(\boldsymbol{\theta}^{(t+1)}))) \right] \leq & R(\mathcal{L}(h(\boldsymbol{\theta}^{(t)}))) - \eta \langle \nabla_{\boldsymbol{\theta}}(R \circ \mathcal{L} \circ h)(\boldsymbol{\theta}^{(t)}), \sum_{i=1}^n \bar{p}_i(\hat{\mathcal{L}}) \nabla_{\boldsymbol{\theta}}(\underset{i}{\mathcal{L}} \circ h_i)(\boldsymbol{\theta}^{(t)}) \rangle \\
& + \eta \left\| \nabla_h (R \circ \mathcal{L})(h(\boldsymbol{\theta}^{(t)})) \right\|_{1,2} O\left( \frac{L^2 \omega^{1/3} \sqrt{nm \log(m)}}{\sqrt{d}} \right) \sqrt{\sum_{i=1}^n \left\| \bar{p}_i(\hat{\mathcal{L}}) \nabla_{h_i} \underset{i}{\mathcal{L}}(h_i(\boldsymbol{\theta}^{(t)})) \right\|_2^2} \\
& + \eta^2 O\left( \left( \frac{\beta C(\mathcal{L})^2}{n\rho} + 2C(\nabla \mathcal{L}) \right) \frac{nL^2 m}{d} \right) O\left( \frac{m}{d\alpha} \right) \sum_{i=1}^n \left\| \bar{p}_i(\hat{\mathcal{L}}) \nabla_{h_i} \underset{i}{\mathcal{L}}(h_i(\boldsymbol{\theta}^{(t)})) \right\|_2^2
\end{aligned}
\tag{88}
$$

where the differences with respect to (72) comes from the fact that $\hat{\mathcal{L}}$ is not the exact loss history here, i.e. $\hat{\mathcal{L}} \neq \check{\mathcal{L}}$, which leads to

$$
\begin{aligned}
\nabla_{\boldsymbol{\theta}}(R \circ \mathcal{L} \circ h)(\boldsymbol{\theta}^{(t)}) &= \sum_{i=1}^n \hat{p}_i(\check{\mathcal{L}}) \nabla_{\boldsymbol{\theta}}(\underset{i}{\mathcal{L}} \circ h_i)(\boldsymbol{\theta}^{(t)}) \\
&\neq \sum_{i=1}^n \bar{p}_i(\hat{\mathcal{L}}) \nabla_{\boldsymbol{\theta}}(\underset{i}{\mathcal{L}} \circ h_i)(\boldsymbol{\theta}^{(t)})
\end{aligned}
\tag{89}
$$

And

$$
\begin{aligned}
\left\| \nabla_h (R \circ \mathcal{L})(h(\boldsymbol{\theta}^{(t)})) \right\|_{1,2} &= \sum_{i=1}^n \left\| \hat{p}_i(\check{\mathcal{L}}) \nabla_{h_i} \underset{i}{\mathcal{L}}(h_i(\boldsymbol{\theta}^{(t)})) \right\|_2 \\
&\neq \sum_{i=1}^n \left\| \hat{p}_i(\hat{\mathcal{L}}) \nabla_{h_i} \underset{i}{\mathcal{L}}(h_i(\boldsymbol{\theta}^{(t)})) \right\|_2
\end{aligned}
\tag{90}
$$

Let

$$K' = C(\mathcal{L}) A(\nabla \mathcal{L}) O\left( \frac{\beta L m^{3/2} \log^2(m)}{\alpha n^{1/2} \rho d^{3/2} b \log\left(\frac{1}{1-\alpha}\right)} \right) \tag{91}$$

Where $C(\mathcal{L}) > 0$ is a constant such that $\mathcal{L}$ is $C(\mathcal{L})$-Lipschitz continuous, and $A(\nabla \mathcal{L}) > 0$ is a constant that bound the gradient of $\mathcal{L}$ with respect to its input.

$C(\mathcal{L})$ and $A(\nabla \mathcal{L})$ are guaranteed to exist under assumptions B.1.

We can prove that, with probability at least $1 - \exp\left(-\Omega\left(\log^2(m)\right)\right)$,

- according to lemma D.8.1

$$\left\|\hat{p}(\hat{\mathcal{L}}) - \hat{p}(\check{\mathcal{L}})\right\|_2 = \sqrt{\sum_{i=1}^{n}\left(\hat{p}_i(\hat{\mathcal{L}}) - \hat{p}_i(\check{\mathcal{L}})\right)^2} \leq \eta\alpha K' \tag{92}$$

- according to lemma D.8.2

$$\left|\langle\nabla_{\boldsymbol{\theta}}(R \circ \mathcal{L} \circ h)(\boldsymbol{\theta}^{(t)}) - \sum_{i=1}^{n}\bar{p}_i(\hat{\mathcal{L}})\nabla_{\boldsymbol{\theta}}(\underset{i}{\mathcal{L}} \circ h_i)(\boldsymbol{\theta}^{(t)})), \sum_{i=1}^{n}\bar{p}_i(\hat{\mathcal{L}})\nabla_{\boldsymbol{\theta}}(\underset{i}{\mathcal{L}} \circ h_i)(\boldsymbol{\theta}^{(t)}))\rangle\right|$$
$$\leq \eta\frac{m}{d}K'\sum_{i=1}^{n}\left\|\bar{p}_i(\hat{\mathcal{L}})\nabla_{\boldsymbol{\theta}}(\underset{i}{\mathcal{L}} \circ h_i)(\boldsymbol{\theta}^{(t)}))\right\|_2^2 \tag{93}$$

- according to lemma D.8.3

$$\left\|\nabla_h(R \circ \mathcal{L})(h(\boldsymbol{\theta}^{(t)}))\right\|_{1,2} \leq \left(\sqrt{n} + \eta K'\right)\sqrt{\sum_{i=1}^{n}\left\|\bar{p}_i(\hat{\mathcal{L}})\nabla_{\boldsymbol{\theta}}(\underset{i}{\mathcal{L}} \circ h_i)(\boldsymbol{\theta}^{(t)}))\right\|_2^2} \tag{94}$$

Combining those three inequalities with (88) we obtain

$$\mathbb{E}_S\left[R(\mathcal{L}(h(\boldsymbol{\theta}^{(t+1)})))\right] - R(\mathcal{L}(h(\boldsymbol{\theta}^{(t)}))) \leq$$
$$\eta\left[-\Omega\left(\frac{m\delta}{dn^2}\right) + O\left(\frac{nL^2m\omega^{1/3}\sqrt{\log(m)}}{d}\right)\right]\sum_{i=1}^{n}\left\|\bar{p}_i(\hat{\mathcal{L}})\nabla_{h_i}\underset{i}{\mathcal{L}}(h_i(\boldsymbol{\theta}^{(t)}))\right\|_2^2$$
$$\eta^2 O\left(K\frac{(n/\alpha)L^2m^2}{d^2} + \left(1 + \frac{m}{d}\right)K'\right)\sum_{i=1}^{n}\left\|\bar{p}_i(\hat{\mathcal{L}})\nabla_{h_i}\underset{i}{\mathcal{L}}(h_i(\boldsymbol{\theta}^{(t)}))\right\|_2^2 \tag{95}$$

One can see that compared to (72), there is only the additional term $\left(1 + \frac{m}{d}\right)K'$.

Using our choice of $\eta$,

$$\eta = \eta_{stale} \leq O\left(\frac{\delta}{n^2K'}\eta_{exact}\right) \tag{96}$$

where $\eta_{exact}$ is the learning rate of Theorem B.1, we have

$$\Omega\left(\frac{\eta m\delta}{dn^2}\right) \geq O\left(\eta^2\left(1 + \frac{m}{d}\right)K'\right) \tag{97}$$

As a result, $\eta^2\left(1 + \frac{m}{d}\right)K'$ is dominated by the term $\Omega\left(\frac{\eta m\delta}{dn^2}\right)$

In addition, since $\eta_{stale} \leq \eta_{exact}$, $\Omega\left(\frac{\eta m\delta}{dn^2}\right)$ still dominates also the ther terms as in the proof of Theorem B.1.

As a consequence, we obtain as in (76) that for any iteration $t > 0$ (after the end of the warm-up)

$$\mathbb{E}_{S_t}\left[R(\mathcal{L}(h(\boldsymbol{\theta}^{(t+1)})))\right] \leq R(\mathcal{L}(h(\boldsymbol{\theta}^{(t)}))) - \Omega\left(\frac{\eta m\delta}{dn^2}\right)\sum_{i=1}^{n}\left\|\bar{p}_i(\hat{\mathcal{L}})\nabla_{h_i}\underset{i}{\mathcal{L}}(h_i(\boldsymbol{\theta}^{(t)}))\right\|_2^2 \tag{98}$$

This concludes the proof using the same arguments as in the end of the proof of Theorem B.1 starting from (76).

### D.8.1 Proof of technical lemma 2

Using Lemma 4.2 and Lemma 4.1 we obtain

$$
\begin{aligned}
\left\| \hat{p}(\hat{\mathcal{L}}) - \hat{p}(\breve{\mathcal{L}}) \right\|_2 &= \left\| \nabla_v R(\hat{\mathcal{L}}) - \nabla_v R(\breve{\mathcal{L}}) \right\|_2 \\
&\leq \frac{\beta}{n\rho} \left\| \hat{\mathcal{L}} - \breve{\mathcal{L}} \right\|_2
\end{aligned}
\tag{99}
$$

Using assumptions B.2 and (Allen-Zhu et al., 2019, Claim 11.2)

$$
\begin{aligned}
\left\| \hat{p}(\hat{\mathcal{L}}) - \hat{p}(\breve{\mathcal{L}}) \right\|_2 &\leq \frac{\beta}{n\rho} \sqrt{ \sum_{i=1}^{n} \left( \underset{i}{\mathcal{L}} \circ h_i(\boldsymbol{\theta}^{(t)}) - \underset{i}{\mathcal{L}} \circ h_i(\boldsymbol{\theta}^{(t_i(t))}) \right)^2 } \\
&\leq \frac{\beta}{n\rho} C(\mathcal{L}) C(h) \sqrt{ \sum_{i=1}^{n} \left\| \boldsymbol{\theta}^{(t)} - \boldsymbol{\theta}^{(t_i(t))} \right\|_{2,2}^2 } \\
&\leq C(\mathcal{L}) O\left( \frac{\beta L m^{1/2}}{n\rho d^{1/2}} \right) \sqrt{ \sum_{i=1}^{n} \left\| \boldsymbol{\theta}^{(t)} - \boldsymbol{\theta}^{(t_i(t))} \right\|_{2,2}^2 }
\end{aligned}
\tag{100}
$$

Where $C(\mathcal{L})$ is the constant of Lipschitz continuity of the per-example loss $\mathcal{L}$ (see assumptions B.2) and $C(h)$ is the constant of Lipschitz continuity of the deep neural network $h$ with respect to its parameters $\boldsymbol{\theta}$.

By developing the recurrence formula of $\boldsymbol{\theta}^{(t)}$ (85), we obtain

$$
\begin{aligned}
\left\| \hat{p}(\hat{\mathcal{L}}) - \hat{p}(\breve{\mathcal{L}}) \right\|_2 &\leq C(\mathcal{L}) O\left( \frac{\beta L m^{1/2}}{n\rho d^{1/2}} \right) \sqrt{ \sum_{i=1}^{n} \left\| \boldsymbol{\theta}^{(t_i(t))} - \left( \sum_{\tau=t_i(t)}^{t-1} \frac{\eta}{b} \sum_{j \in S_\tau} \nabla_{\boldsymbol{\theta}} (\underset{j}{\mathcal{L}} \circ h_j)(\boldsymbol{\theta}^{(\tau)}) \right) - \boldsymbol{\theta}^{(t_i(t))} \right\|_{2,2}^2 } \\
&\leq \eta C(\mathcal{L}) O\left( \frac{\beta L m^{1/2}}{n\rho d^{1/2}} \right) \sqrt{ \sum_{i=1}^{n} \left\| \sum_{\tau=t_i(t)}^{t-1} \frac{1}{b} \sum_{j \in S_\tau} \nabla_{\boldsymbol{\theta}} (\underset{j}{\mathcal{L}} \circ h_j)(\boldsymbol{\theta}^{(\tau)}) \right\|_{2,2}^2 }
\end{aligned}
$$

Let $A(\nabla \mathcal{L})$ a bound on the gradient of the per-example loss function. Using Theorem D.2 and the chain rule

$$
\forall j, \ \forall \tau \quad \left\| \nabla_{\boldsymbol{\theta}} (\underset{j}{\mathcal{L}} \circ h_j)(\boldsymbol{\theta}^{(\tau)}) \right\|_{2,2} \leq A(\nabla \mathcal{L}) O\left( \frac{m}{d} \right)
\tag{101}
$$

And using the triangular inequality

$$
\begin{aligned}
\left\| \sum_{\tau=t_i(t)}^{t-1} \frac{1}{b} \sum_{j \in S_\tau} \nabla_{\boldsymbol{\theta}} (\underset{j}{\mathcal{L}} \circ h_j)(\boldsymbol{\theta}^{(\tau)}) \right\|_{2,2} &\leq \sum_{\tau=t_i(t)}^{t-1} \frac{1}{b} \sum_{j \in S_\tau} \left\| \nabla_{\boldsymbol{\theta}} (\underset{j}{\mathcal{L}} \circ h_j)(\boldsymbol{\theta}^{(\tau)}) \right\|_{2,2} \\
&\leq \sum_{\tau=t_i(t)}^{t-1} A(\nabla \mathcal{L}) O\left( \frac{m}{d} \right) \\
&\leq A(\nabla \mathcal{L}) O\left( \frac{m}{d} \right) (t - t_i(t))
\end{aligned}
\tag{102}
$$

As a result, we obtain

$$
\left\| \hat{p}(\hat{\mathcal{L}}) - \hat{p}(\breve{\mathcal{L}}) \right\|_2 \leq \eta C(\mathcal{L}) A(\nabla \mathcal{L}) O\left( \frac{\beta L m^{3/2}}{n\rho d^{3/2}} \right) \sqrt{ \sum_{i=1}^{n} (t - t_i(t))^2 }
\tag{103}
$$

For all $i$ and for any $\tau$ the probability that the sample $i$ is not in batch $S_\tau$ is lesser than $(1 - \alpha)^b$.

Therefore, for any $k \geq 1$ and for any $t$,

$$P\left(t - t_i(t) \geq k\right) \leq (1-\alpha)^{kb} \tag{104}$$

For $k \geq \frac{1}{b}\Omega\left(\frac{\log^2(m)}{\log\left(\frac{1}{1-\alpha}\right)}\right)$, we have $(1-\alpha)^{kb} \leq \exp\left(-\Omega\left(\log^2(m)\right)\right)$, and thus with probability at least $1 - \exp\left(-\Omega\left(\log^2(m)\right)\right)$,

$$\forall t, \quad t - t_i(t) \leq O\left(\frac{\log^2(m)}{b\log\left(\frac{1}{1-\alpha}\right)}\right) \tag{105}$$

As a result, we finally obtain that with probability at least $1 - \exp\left(-\Omega\left(\log^2(m)\right)\right)$,

$$
\begin{aligned}
\left\|\hat{p}(\hat{\mathcal{L}}) - \hat{p}(\check{\mathcal{L}})\right\|_2 &\leq \eta C(\mathcal{L}) A(\nabla \mathcal{L}) O\left(\frac{\beta L m^{3/2}}{n\rho d^{3/2}}\right) \sqrt{n} O\left(\frac{\log^2(m)}{b\log\left(\frac{1}{1-\alpha}\right)}\right) \\
&\leq \eta\alpha O\left(\frac{\beta L m^{3/2}\log^2(m)}{\alpha n^{1/2}\rho d^{3/2}b\log\left(\frac{1}{1-\alpha}\right)}\right) \\
&\leq \eta\alpha K'
\end{aligned}
\tag{106}
$$

### D.8.2 PROOF OF TECHNICAL LEMMA 3

Let us first denote

$$
\begin{aligned}
A &= \left|\langle\nabla_{\boldsymbol{\theta}}(R\circ\mathcal{L}\circ h)(\boldsymbol{\theta}^{(t)}) - \sum_{i=1}^n \bar{p}_i(\hat{\mathcal{L}})\nabla_{\boldsymbol{\theta}}(\underset{i}{\mathcal{L}}\circ h_i)(\boldsymbol{\theta}^{(t)})), \sum_{i=1}^n \bar{p}_i(\hat{\mathcal{L}})\nabla_{\boldsymbol{\theta}}(\underset{i}{\mathcal{L}}\circ h_i)(\boldsymbol{\theta}^{(t)}))\rangle\right| \\
&= \left|\langle\sum_{i=1}^n \left(\bar{p}_i(\check{\mathcal{L}}) - \bar{p}_i(\hat{\mathcal{L}})\right)\nabla_{\boldsymbol{\theta}}(\underset{i}{\mathcal{L}}\circ h_i)(\boldsymbol{\theta}^{(t)})), \sum_{i=1}^n \bar{p}_i(\hat{\mathcal{L}})\nabla_{\boldsymbol{\theta}}(\underset{i}{\mathcal{L}}\circ h_i)(\boldsymbol{\theta}^{(t)}))\rangle\right|
\end{aligned}
\tag{107}
$$

Using Cauchy-Schwarz inequality

$$
\begin{aligned}
A &= \left|\sum_{i=1}^n \left(\bar{p}_i(\check{\mathcal{L}}) - \bar{p}_i(\hat{\mathcal{L}})\right)\langle\nabla_{\boldsymbol{\theta}}(\underset{i}{\mathcal{L}}\circ h_i)(\boldsymbol{\theta}^{(t)})), \sum_{j=1}^n \bar{p}_j(\hat{\mathcal{L}})\nabla_{\boldsymbol{\theta}}(\underset{j}{\mathcal{L}}\circ h_j)(\boldsymbol{\theta}^{(t)}))\rangle\right| \\
&\leq \left\|\hat{p}(\hat{\mathcal{L}}) - \hat{p}(\check{\mathcal{L}})\right\|_2 \sqrt{\sum_{i=1}^n \left(\langle\nabla_{\boldsymbol{\theta}}(\underset{i}{\mathcal{L}}\circ h_i)(\boldsymbol{\theta}^{(t)})), \sum_{j=1}^n \bar{p}_j(\hat{\mathcal{L}})\nabla_{\boldsymbol{\theta}}(\underset{j}{\mathcal{L}}\circ h_j)(\boldsymbol{\theta}^{(t)}))\rangle\right)^2}
\end{aligned}
\tag{108}
$$

Let

$$B = \langle\nabla_{\boldsymbol{\theta}}(\underset{i}{\mathcal{L}}\circ h_i)(\boldsymbol{\theta}^{(t)})), \sum_{j=1}^n \bar{p}_j(\hat{\mathcal{L}})\nabla_{\boldsymbol{\theta}}(\underset{j}{\mathcal{L}}\circ h_j)(\boldsymbol{\theta}^{(t)}))\rangle \tag{109}$$

Using again Cauchy-Schwarz inequality

$$B \leq \left\|\nabla_{\boldsymbol{\theta}}(\underset{i}{\mathcal{L}}\circ h_i)(\boldsymbol{\theta}^{(t)}))\right\|_{2,2} \left\|\sum_{j=1}^n \bar{p}_j(\hat{\mathcal{L}})\nabla_{\boldsymbol{\theta}}(\underset{j}{\mathcal{L}}\circ h_j)(\boldsymbol{\theta}^{(t)}))\right\|_{2,2} \tag{110}$$

As a result, $A$ becomes

$$
\begin{aligned}
A &\leq \left\| \hat{p}(\hat{\mathcal{L}}) - \hat{p}(\breve{\mathcal{L}}) \right\|_2 \left\| \sum_{j=1}^n \bar{p}_j(\hat{\mathcal{L}}) \nabla_{\boldsymbol{\theta}} (\underset{j}{\mathcal{L}} \circ h_j)(\boldsymbol{\theta}^{(t)}) \right\|_{2,2} \sqrt{\sum_{i=1}^n \left\| \nabla_{\boldsymbol{\theta}} (\underset{i}{\mathcal{L}} \circ h_i)(\boldsymbol{\theta}^{(t)}) \right\|_{2,2}^2} \\
&\leq \left\| \hat{p}(\hat{\mathcal{L}}) - \hat{p}(\breve{\mathcal{L}}) \right\|_2 \left\| \sum_{j=1}^n \bar{p}_j(\hat{\mathcal{L}}) \nabla_{\boldsymbol{\theta}} (\underset{j}{\mathcal{L}} \circ h_j)(\boldsymbol{\theta}^{(t)}) \right\|_{2,2} \sqrt{\sum_{i=1}^n \frac{1}{\alpha^2} \left\| \bar{p}_j(\hat{\mathcal{L}}) \nabla_{\boldsymbol{\theta}} (\underset{i}{\mathcal{L}} \circ h_i)(\boldsymbol{\theta}^{(t)}) \right\|_{2,2}^2} \\
&\leq \frac{1}{\alpha} \left\| \hat{p}(\hat{\mathcal{L}}) - \hat{p}(\breve{\mathcal{L}}) \right\|_2 \left\| \sum_{j=1}^n \bar{p}_j(\hat{\mathcal{L}}) \nabla_{\boldsymbol{\theta}} (\underset{j}{\mathcal{L}} \circ h_j)(\boldsymbol{\theta}^{(t)}) \right\|_{2,2}^2
\end{aligned}
\tag{111}
$$

Using the triangular inequality, Theorem D.2, and Lemma D.8.1, we finally obtain

$$
\begin{aligned}
A &\leq \frac{m}{\alpha d} \left\| \hat{p}(\hat{\mathcal{L}}) - \hat{p}(\breve{\mathcal{L}}) \right\|_2 \sum_{j=1}^n \left\| \bar{p}_j(\hat{\mathcal{L}}) \nabla_{h_j} \underset{j}{\mathcal{L}}(h_j(\boldsymbol{\theta}^{(t)})) \right\|_{2,2}^2 \\
&\leq \eta \frac{m}{d} K' \sum_{j=1}^n \left\| \bar{p}_j(\hat{\mathcal{L}}) \nabla_{h_j} \underset{j}{\mathcal{L}}(h_j(\boldsymbol{\theta}^{(t)})) \right\|_{2,2}^2
\end{aligned}
\tag{112}
$$

### D.8.3 PROOF OF TECHNICAL LEMMA 4

We have

$$
\begin{aligned}
\left\| \nabla_h (R \circ \mathcal{L})(h(\boldsymbol{\theta}^{(t)})) \right\|_{1,2} &= \sum_{j=1}^n \bar{p}_j(\breve{\mathcal{L}}) \left\| \nabla_{h_j} \underset{j}{\mathcal{L}}(h_j(\boldsymbol{\theta}^{(t)})) \right\|_{2,2} \\
&= \sum_{j=1}^n \bar{p}_j(\hat{\mathcal{L}}) \left\| \nabla_{h_j} \underset{j}{\mathcal{L}}(h_j(\boldsymbol{\theta}^{(t)})) \right\|_{2,2} \\
&+ \sum_{j=1}^n \left( \frac{\bar{p}_j(\breve{\mathcal{L}}) - \bar{p}_j(\hat{\mathcal{L}})}{\bar{p}_j(\hat{\mathcal{L}})} \right) \bar{p}_j(\hat{\mathcal{L}}) \left\| \nabla_{h_j} \underset{j}{\mathcal{L}}(h_j(\boldsymbol{\theta}^{(t)})) \right\|_{2,2}
\end{aligned}
\tag{113}
$$

Using Cauchy-Schwarz inequality

$$
\left\| \nabla_h (R \circ \mathcal{L})(h(\boldsymbol{\theta}^{(t)})) \right\|_{1,2} \leq \left( \sqrt{n} + \sqrt{\sum_{j=1}^n \left( \frac{\bar{p}_j(\breve{\mathcal{L}}) - \bar{p}_j(\hat{\mathcal{L}})}{\bar{p}_j(\hat{\mathcal{L}})} \right)^2} \right) \sqrt{\sum_{j=1}^n \left\| \bar{p}_j(\hat{\mathcal{L}}) \nabla_{h_j} \underset{j}{\mathcal{L}}(h_j(\boldsymbol{\theta}^{(t)})) \right\|_{2,2}^2}
\tag{114}
$$

Using Lemma D.8.1

$$
\begin{aligned}
\sum_{j=1}^n \left( \frac{\bar{p}_j(\breve{\mathcal{L}}) - \bar{p}_j(\hat{\mathcal{L}})}{\bar{p}_j(\hat{\mathcal{L}})} \right)^2 &\leq \frac{1}{\alpha} \left\| \hat{p}(\hat{\mathcal{L}}) - \hat{p}(\breve{\mathcal{L}}) \right\|_2 \\
&\leq \eta K'
\end{aligned}
\tag{115}
$$

Therefore, we finally obtain

$$
\left\| \nabla_h (R \circ \mathcal{L})(h(\boldsymbol{\theta}^{(t)})) \right\|_{1,2} \leq \left( \sqrt{n} + \eta K' \right) \sqrt{\sum_{j=1}^n \left\| \bar{p}_j(\hat{\mathcal{L}}) \nabla_{h_j} \underset{j}{\mathcal{L}}(h_j(\boldsymbol{\theta}^{(t)})) \right\|_{2,2}^2}
\tag{116}
$$

