# OpenReview forum: "SGD with Hardness Weighted Sampling for Distributionally Robust Deep Learning"
_ICLR.cc/2020/Conference — Reject_

### Official Review · AnonReviewer4 · 2019-10-30
**Official Blind Review #4**

**Rating:** 3

**Review:**

This paper proposes a method for Distributionally Robust Optimization (DRO). DRO has recently been proposed (Namkoong and Duchi 2016 and others) as a robust learning framework compared to Empirical Risk Minimization (ERM).  My analysis of this work in short is that the problem that this paper addresses is interesting and not yet solved. This paper proposes a simple and efficient method for DRO. However, convergence guarantees are weak which is reflected in their weak experimental results. And for a 10-page paper, the writing has to improve quite a lot.

Summary of contributions:
The phi-divergence variation of DRO introduces a min-max objective that minimizes a maximally reweighted empirical risk. The main contribution of this work is as follows:
- Show that for two common phi-divergences (KL-divergence and Pearson-chi2 divergence), the inner maximization has a simple solution for the weights that depends on the value of the loss on the training set.
- Use the above algorithm and modify SGD by changing the sampling of data according to the weights.
- Convergence proofs for the above algorithm for wide networks (not necessarily infinite-width).


Cons:
- Fig 1, ERM should not be so bad as if predicting randomly. This suggests a problem in the experimental setting. At least more ablation study is needed, e.g. try varying the percentage of data kept and show the final test accuracy as a function of this percentage for ERM. In the worst case, the accuracy for ERM should be ~90% on 9 classes and zero on 10th which is ~80% for uniform test accuracy. Another point, what is the train accuracy on cats? ERM should be overfitting to those few cats in the training set and achieve some non-zero accuracy at test. But it seems the test accuracy is almost exactly 0 at test time.
- In Theorem 5.1 that provides convergence proofs, the learning rate eta_exact has a dependence on 1/beta, which means much smaller learning rates should be used for the proposed method. However, in the experiments it seems that the same learning rate is used for all methods. This seems to trouble the convergence very clearly as the curves start to fluctuate considerably by increasing for larger betas. There is no bounds on beta which in practice can force us to use orders of magnitude smaller learning rates.
- Section 4.3 aims at linking hard negative mining to the proposed method. What they actually do is propose a new definition for hard-negative mining which is satisfied by the proposed method. There is little basis for suggesting this definition. There are myriads of work on hard-negative mining and suggesting a new definition needs a more thorough study of related works.
- Section A.1.1 argues we would stop ERM when it plateaus, but it doesn't look like it has plateaued in fig 2 left.
- Section A.3, I'm not sure lr=1 would be stable wide resnet. Maybe there is a problem with the experimental setting.
- There is such much non-crucial details in the main body that increased the main text to 10 pages. At least 2 pages could be saved by moving details of theorems and convergence results to the appendix.

After rebuttal:
I'm keeping more score. The manuscript has been improved quite a lot.  My main concern remains the experiments. Both texts and mathematical statements still need more edits.

The authors have completely removed experiments on cifar10. Paper now has only one set of experiments on MNIST with little ablation study. I still think my suggested ablation study is interesting. As a minimal sanity check, I'd like to see the performance of DRO on the original MNIST dataset. Basically, how much do we lose by using a robust method on a balanced dataset? Even with full ablation studies, I don't think MNIST is not enough for testing methods based on hard-negative mining.

**Experience Assessment:**

I have published one or two papers in this area.

**Review Assessment: Checking Correctness Of Derivations And Theory:**

I assessed the sensibility of the derivations and theory.

**Review Assessment: Checking Correctness Of Experiments:**

I assessed the sensibility of the experiments.

**Review Assessment: Thoroughness In Paper Reading:**

I read the paper at least twice and used my best judgement in assessing the paper.

---

> ### Author Response · Authors · 2019-11-08
> **First feedbacks**
>
> Thank you for taking the time to review our submission and for your insights.
>
> In this first answer to your review, to foster rapid discussion iterations, we feedback on some of your concerns below. We will come back to you shortly regarding your other concerns, as we are running additional experiments and working on the manuscript.
>
> [Definition of Hard Example Mining Sampling]
> To the best of our knowledge, there is no widely accepted definition for Hard Example Mining in the literature. Yet, we need a definition to study the link between DRO and Hard Example Mining. We do not pretend to propose a definition for all possible Hard Example Mining methods, which is why we used the term “Hard Example Mining Sampling” rather than only “Hard Example Mining”.
>
> Our definition further address only Online Hard Example Mining as defined in [1] since there is no discrete switching between training and hard examples mining. To make it clearer that our definition corresponds only to a subset of Hard Example Mining methods, we have now renamed it as “Online Hard Example Mining Sampling” in the revised version of the submission.
>
> [The learning rate is too high for the large values of beta]
> Please see our common comment about our experiments on MNIST.
>
> [When does ERM plateaus in fig.2 left?]
> Please see our common comment about our experiments on MNIST.
>
> [Reducing the size of the main text]
> We are working on moving the details of section 5 to the appendix as you suggested to improve clarity.
>
>
> References:
> [1] Training Region-based Object Detectors with Online Hard Example Mining. A. Srivastava et al. 2016

---

> ### Author Response · Authors · 2019-11-14
> **CIFAR10 experiment and Writing**
>
> [Writing]
> We have worked on the writing and have now reduced the main text to 8 pages by moving the details of the convergence theorems to the appendix as you suggested.
>
> [CIFAR10]
> Please see our common reply "Experiments on CIFAR10 and DRO with momentum".
> Thank you.

---

### Official Review · AnonReviewer2 · 2019-11-04
**Official Blind Review #2**

**Rating:** 3

**Review:**

This paper studies the Distributionally Robust Optimization (DRO), in the sense that the weights assigned to the training data can change, but the training data itself remains unchanged. They demonstrate that SGD with hardness weighted sampling is a principled and efficient optimization method for DRO in machine learning and is particularly suited in the context of deep learning. On the theoretical side, they prove the convergence of our DRO algorithm for over-parameterized
deep learning networks with ReLU activation and finite number of layers and parameters.

The DRO problem studied in this paper is relatively easy, in the sense that the inner maximization has close form solution (at least for KL divergence). Therefore, the proposed method is straightforward. All the derivations in Section 3 and 4 are strainghtforward and sensible. I do not know any paper that has proposed Algorithm 1 (or something similar) before, but I will be surprised there is not. I do not check the derivations in Section 5, but I suppose that it is a small modification of the proof in Allen-Zhu et al., 2019. I suppose that the theorem is correct, but it provides nearly no practical guidance.

In Algorithm 1 Line 18-19, the algorithm proposes to do sampling with replacement using the softmax probability \hat{p}. How about directly multipling the weights \hat{p} to the current minibatch of samples. (i.e., re-weighting the samples instead of re-sampling the samples). What's the difference between these two choices?

Second, both experiments are not convincing enough. In the CIFAR10 experiment (Figure 1), the baseline method ERM is too low. I guess that this low number is due to the large initial learning rate 1. The authors should provide the best performance IRM can achieve, and compare with the best performance DRO can achieve. Although the authors are claiming that the proposed DRO works with larger learning rate, Figure 1 (Left) simply gives readers the wrong information that ERM does not work. "Figure 2 suggests that if we train ERM long enough it will converge to the same accuracy as DRO." The objectives of ERM and DRO are different. Their accuracy may become closer to each other, but I'm not sure they will finally achieve the same accuracy (the same optimum). Can the authors elaborate on this?

Finally, the focal loss has achieved good empirical results in object detection. I feel that it can be formulated as a special case of the Equation (2) and Algorithm 1, by picking up some proper \phi-divergence and using the (un-normalized) re-weighting scheme. It will good if the authors can provide a principled view of focal loss through the lens of DRO.


**Experience Assessment:**

I have read many papers in this area.

**Review Assessment: Checking Correctness Of Derivations And Theory:**

I assessed the sensibility of the derivations and theory.

**Review Assessment: Checking Correctness Of Experiments:**

I assessed the sensibility of the experiments.

**Review Assessment: Thoroughness In Paper Reading:**

I read the paper at least twice and used my best judgement in assessing the paper.

---

> ### Author Response · Authors · 2019-11-08
> **First feedbacks**
>
> Thank you for taking the time to review our submission and for your insights.
>
> In this first answer to your review, to foster rapid discussion iterations, we feedback on some of your concerns below. We will come back to you shortly regarding your other concerns, as we are running additional experiments and working on the manuscript.
>
> [On the novelty of the paper]
> To the best of our knowledge, our paper is the first to propose a weighted sampling strategy for deep DRO. In addition to offer a pragmatic algorithm, we also mathematically prove the convergence of our approach. Research on DRO in the non-convex setting has only started to appear recently. The differences between existing recent works and our algorithm are discussed in section 4 of our submission. The timeliness of our work is further confirmed by the fact that another work on a similar topic was independently submitted at ICLR this year [1].
>
> [Clarification following your comment: “The DRO problem studied in this paper is relatively easy, in the sense that the inner maximization has close form solution”]
> It is correct that there is a closed form solution for the inner maximization problem, provided we have access to the full exact per-sample loss history.  However, this is not possible to evaluate efficiently in practice. To address this limitation, we propose to exploit a stale loss history that is continuously updated. One of our main contributions is to show that this approach is not only intuitive but also leads to provable convergence.
>
> [Contributions compared to Allen-Zhu et al. 2019]
> There were several important technical challenges that made the extension of the result of Allen-Zhu et al. 2019 from ERM to our algorithm for DRO non-trivial. We used the framework of Allen-Zhu as a backbone for our proofs, but in contrast to the ERM case of Allen-Zhu:
>     - The DRO loss is not linear with respect to the per-sample loss because of the max operator. This linearity property is used many times in the proofs of Allen-Zhu et al. 2019.
>     - The sampling distribution is in our case dynamic and not uniform.
>     - Another substantial challenge is the fact that our sampling used with the stale loss only approximates the closed form solution of the internal max problem of the DRO problem.
>
> Addressing these points required substantial work, as can be found in appendix C.6 and C.7 (pages 21-33 in the first version of the submission)
>
>
> [Question 1: about the link between dynamic sampling (our approach) and re-weighting]
> Reweighting while sampling according to the uniform distribution is of course possible. This would correspond to using importance sampling (up to a multiplicative factor n=number of training samples).
> Importance sampling is often used when we cannot sample with respect to the desired distribution or to reduce the variance of the stochastic estimation of the gradient [2,3].
>
> In our case, the proposed hardness weighted sampler is a better approximation of the target distribution than the uniform distribution, which motivates its use.
>
>
> [Question 2: about training longer in the MNIST experiment]
> Please see our common comment about our experiments on MNIST.
>
> [Question 3: about a link between the Focal Loss and DRO]
> It is true that both our hardness weighted sampler and the Focal Loss are motivated by hard examples mining. However, the weights used in the Focal Loss were defined heuristically. In addition, those weights are specific to the cross entropy, while our optimization method can be used with any loss function.
>
> Even though the weights of the Focal Loss can be expressed as a function of the cross entropy loss, It is not clear how these weights can be justified a posteriori as the result of the inner maximization problem of DRO with the cross entropy loss and a given phi-divergence. In particular, the vector of the weights of the Focal Loss is not a probability vector. Therefore, it does not trivially belong to the space of admissible solutions for the inner maximization optimization problem of DRO. More work would be required to establish if deeper connections exists between cross-entropy based DRO and the Focal Loss.
>
>
> References:
> [1] Distributionally Robust Neural Networks. (under submission at ICLR 2020. Paper1796). https://openreview.net/forum?id=ryxGuJrFvS&noteId=BJexEXB09r
>
> [2] An introduction to graphical models. MI Jordan, 2005.
> http://www.cs.cmu.edu/~lebanon/pub/book/
>
> [3] Methods of Reducing Sample Size in Monte Carlo Computations. H. Khan et al, 1953.

---

> ### Author Response · Authors · 2019-11-14
> **Experiment on CIFAR10**
>
> Please see our common reply "Experiments on CIFAR10 and DRO with momentum".
> Thank you.

---

### Official Review · AnonReviewer3 · 2019-11-09
**Official Blind Review #3**

**Rating:** 3

**Review:**

Disclaimer: I was able to read the other reviews and the author’s responses before finalizing this review.

The paper proposes an easy-to-implement algorithm for DRO on example weights, with the same computational cost as SGD.  The algorithm is based on hardness weighted sampling and links are shown to hard example mining.  Convergence results are shown for (finitely) wide networks.  Additionally, the paper claims to demonstrate the usefulness of the algorithm in deep learning.

I am unable to assess how the convergence results place in the literature, but I believe they motivate the algorithm.  The claims of practical usefulness in deep learning do not seem supported by the provided empirical evidence.
For the CIFAR experiments, the ERM baseline does not seem to train - probably due to the choice of learning rate.  This makes it unclear how the proposed algorithm compares.  They claim the algorithm is more robust to the learning rate, so is it possible to train with the original learning rate used for the ERM baseline?  Why was a different learning rate chosen?

If the experimental results showed an improvement over a properly trained ERM baseline on CIFAR I would lean toward weak accept.

Many state-of-the-art deep learning pipelines do not use plain SGD - for example, the WideResNet you used on CIFAR. How is Algorithm 1 used on these? Do we only make changes to the update on line 24?  Using momentum with nested optimization can introduce instabilities. Perhaps combining momentum with nested optimization of loss weights and parameters is why the baseline does not train?  Maybe you could try an architecture that trained using vanilla SGD, so you can better leverage your theoretical results?

It would provide evidence of the usefulness of the algorithm if we could take various pipelines and just drop their optimizer updates into algorithm 1 - hopefully without having to spend time re-tuning their optimizer parameters. It would also be nice to see the training loss/accuracy - perhaps over all classes and just cats - in the appendix.


Things to improve that did not impact the score:

(Page 1) “in term” -> “in terms”
(Page 1 & 2) “an history” -> “a history”
(Page 2) “Optmization” -> “Optimization”
(Page 5) “allows to link” -> “allows us to link”
(Page 7) “we focus at” -> “we focus on”
(Page 8) “allows to guarantee” -> “allows us to guarantee”
(Page 19) “continous” -> “continuous

**Experience Assessment:**

I do not know much about this area.

**Review Assessment: Checking Correctness Of Derivations And Theory:**

I assessed the sensibility of the derivations and theory.

**Review Assessment: Checking Correctness Of Experiments:**

I assessed the sensibility of the experiments.

**Review Assessment: Thoroughness In Paper Reading:**

I read the paper at least twice and used my best judgement in assessing the paper.

---

> ### Author Response · Authors · 2019-11-14
> **Reply to your comments**
>
> Thank you for taking the time to review our submission and for your insights.
>
> We have grouped the reply to your questions on the momentum and our reply to the common concern of the reviewers regarding our results on CIFAR10 in the common reply "Experiments on CIFAR10 and DRO with momentum".

---

### Author Response · Authors · 2019-11-08
**Experiments on MNIST**

We thank the reviewers for taking the time to review our submission and for their insights.

In this comment, we feedback on the concerns of the reviewers regarding our experiments on MNIST.

[Training longer in the MNIST experiment]
We have expanded our experiments with three times more training epochs as per the reviewer suggestions (please see the updated Fig. 2 in the revised submission).

As suggested by AnonReviewer2 “I'm not sure they will finally achieve the same accuracy (the same optimum)”, the updated Fig. 2 suggests that after a large number of iterations, ERM and DRO converge to different local minima, and that DRO leads to a better generalization on the under-represented class.

[On Early-stopping: AnonReviewer4 “Section A.1.1 argues we would stop ERM when it plateaus, but it doesn't look like it has plateaued in fig 2 left.”]
We agree that our statement could be made more accurate. When we used the term “plateau” it was related to the choice of the patience parameter in early-stopping. In the left panel of Fig.2, there is no improvement of ERM of more than 0.04% in accuracy between epoch 20 and 30. For a patience lesser than 10 epochs, one would stop training for ERM before epoch 30. To clarify this point, we rephrased or statement as: “if the patience parameter is too low, ERM might be considered to plateau at an epoch at which it has an accuracy of 0 on the under-represented class”.

It is also worth noting, that this experiment corresponds to the best-case scenario in which the testing set is used for choosing the optimal number of epochs (i.e. there is no bias at all between the validation and the testing distributions).

[On instabilities in the learning curve on the testing set]
AnonReviewer4 suggested that the instabilities observed during the first epochs in the learning curves on the testing set for large values of beta could be due to a value of the learning rate that is too large: “in the experiments it seems that the same learning rate is used for all methods. This seems to trouble the convergence very clearly as the curves start to fluctuate considerably by increasing for larger betas.”

Following this suggestion, we tried to divide the learning rate by 10 or 100. However, we observed no reduction of the instabilities on the testing set.

Looking at the learning curves **on the training set**, we found that the loss curves for beta=10 were actually stable there. However, we have observed that during the iterations for which instabilities appears **on the testing set**, the standard deviation of the loss **on the training set** is relatively high (i.e. the hardness weighted probability is further away from the uniform distribution).

This suggests that the apparent instabilities **on the testing set** are not related to a too high learning rate, but to differences between the DRO loss and the ERM loss.

Following this observation, we increased beta to 100 for the same learning rate and it led to higher accuracy on the testing set (see the updated fig. 2).

---

### Author Response · Authors · 2019-11-14
**Experiments on CIFAR10 and DRO with momentum**

In this comment we reply to the concerns of the reviewers regarding the experiment on CIFAR10.
We also feedback on the concern of AnonReviewer3 regarding the introduction of momentum in the proposed algorithm.

[CIFAR10 training issues for the baseline]
The baseline approach [1] shuffles the training dataset at each epoch. By default, it uses the last batch as is even if it is much smaller than the standard batch size. For the CIFAR10 training set with 10% of the cats, it turned out that the last batch was very small, which in turn led to unstable training.

After changing this behavior so as to drop the last batch, the baseline now trains more smoothly even with the large learning rate used in our experiments.

As such, even though figure 1 of our first submission showed an accurate depiction of the behavior of the baseline method using all the default parameters except for the learning rate, the effect measured was not related to our method. The difference mostly came from the noisy gradients occurring from the small last batch. As a result, this experiment has been removed from the manuscript.

[Momentum]
In the baseline approach [1], the best performance is obtained with SGD with momentum and a learning rate decay schedule.

Following preliminary experiments, we have found that using momentum updates with DRO is not straightforward. We think that this is due to the inner maximization of DRO (as suggested by AnonReviewer3), and that specific momentum updates for DRO are needed.

For large values of beta (beta>10) the WideResNet_28_10 with DRO does not train properly with momentum, except if we reduce the learning rate by two orders of magnitude. In the latter case, it trains very slowly and cannot be properly tested given the duration of the rebuttal period. It is possible that tuning the momentum parameter for DRO could tackle this issue, but we have not yet had the opportunity to evaluate it.

For low values of beta (beta<=1), we can train the WideResNet_28_10 with DRO with momentum and using the same hyperparameters as the baseline without DRO. With such small betas, we obtain the same accuracy as the baseline since the distributionally robust loss is very close to the mean loss (ERM case).

Investigating efficient means of introducing momentum within DRO requires further work. We have commented on this in the conclusion.


[1] Wide residual networks. Zagoruyko, S., & Komodakis, N. (2016).

---

### Decision · Program_Chairs · 2019-12-19

**Decision:**

Reject

**Comment:**

This paper proposes a modification of SGD to do distributionally-robust optimization of deep networks.  The main idea is sensible enough, however, the inadequate handling of baselines and relatively toy nature of the experiments means that this paper needs more work to be accepted.